# Health outcomes after myocardial infarction: A population study of 56 million people in England

Marlous Hall[1,2]*, Lesley Smith[3], Jianhua Wu[2,4], Chris Hayward[1,2], Jonathan A. Batty[1,2], Paul C. Lambert[5,6], Harry Hemingway[7,8,9,10], Chris P. Gale[1,2,11]

1 Leeds Institute of Cardiovascular and Metabolic Medicine, University of Leeds, Leeds, United Kingdom, 2 Leeds Institute for Data Analytics, University of Leeds, Leeds, United Kingdom, 3 Leeds Institute for Health Sciences, University of Leeds, Leeds, United Kingdom, 4 Wolfson Institute of Population Health, Queen Mary University of London, London, United Kingdom, 5 Biostatistics Research Group, Department of Population Health Sciences, University of Leicester, Leicester, United Kingdom, 6 Department of Medical Epidemiology and Biostatistics, Karolinska Institutet, Stockholm, Sweden, 7 Institute of Health Informatics, University College London, London, United Kingdom, 8 Health Data Research UK, University College London, London, United Kingdom, 9 NIHR Biomedical Research Centre, University College London Hospitals NHS Foundation Trust, University College London, London, United Kingdom, 10 Charité Universitätsmedizin, Berlin, Germany, 11 Department of Cardiology, Leeds Teaching Hospitals NHS Trust, Leeds, United Kingdom

* m.s.hall@leeds.ac.uk

**Data Availability Statement:** The data underlying the results presented in the study are available from NHS England's Data Access Request Service https://dataaccessrequest.hscic.gov.uk/. HES data

## Abstract

### Background

The occurrence of a range of health outcomes following myocardial infarction (MI) is unknown. Therefore, this study aimed to determine the long-term risk of major health outcomes following MI and generate sociodemographic stratified risk charts in order to inform care recommendations in the post-MI period and underpin shared decision making.

### Methods and findings

This nationwide cohort study includes all individuals aged ≥18 years admitted to one of 229 National Health Service (NHS) Trusts in England between 1 January 2008 and 31 January 2017 (final follow-up 27 March 2017). We analysed 11 non-fatal health outcomes (subsequent MI and first hospitalisation for heart failure, atrial fibrillation, cerebrovascular disease, peripheral arterial disease, severe bleeding, renal failure, diabetes mellitus, dementia, depression, and cancer) and all-cause mortality. Of the 55,619,430 population of England, 34,116,257 individuals contributing to 145,912,852 hospitalisations were included (mean age 41.7 years (standard deviation [SD 26.1]); n = 14,747,198 (44.2%) male). There were 433,361 individuals with MI (mean age 67.4 years [SD 14.4)]; n = 283,742 (65.5%) male). Following MI, all-cause mortality was the most frequent event (adjusted cumulative incidence at 9 years 37.8% (95% confidence interval [CI] [37.6,37.9]), followed by heart failure (29.6%; 95% CI [29.4,29.7]), renal failure (27.2%; 95% CI [27.0,27.4]), atrial fibrillation (22.3%; 95% CI [22.2,22.5]), severe bleeding (19.0%; 95% CI [18.8,19.1]), diabetes (17.0%; 95% CI [16.9,17.1]), cancer (13.5%; 95% CI [13.3,13.6]), cerebrovascular disease (12.5%; 95% CI [12.4,12.7]), depression (8.9%; 95% CI [8.7,9.0]), dementia (7.8%; 95% CI

are managed and released by NHS Digital. The specific extract provided to the research team can only be used for the stated purpose of the study and for the length of time necessary to conduct the study. The extract cannot be shared outside of the research team or for any other purpose according to the legally binding terms under which they were released. Please see our privacy notice for further information on the purpose and legal basis of our use of these data: https://digital.nhs.uk/data-and-information/data-tools-and-services/data-services/hospital-episode-statistics. Access to HES data is available by direct application to NHS Digital and is available to anyone who has a legal basis for accessing these data, meets the requirements for safe and secure use of these data and intends to use these data for demonstrable benefit to health and social care in the UK. A full HES data dictionary, information of how to apply and the costs associated with data applications are available publicly via the NHS digital website: https://digital.nhs.uk. All diagnostic and procedure codes used to define specific study outcomes are provided in the supplementary online material released at time of publication. Aggregated data of the age, sex and deprivation specific post MI absolute risk of new onset disease (as presented in heat maps (Fig 5)) are available to explore freely via: https://multimorbidity-research-leeds.github.io/research-resources Anyone wishing to use these aggregate data to generate their own graphical summaries may do so providing full reference is given to this publication.

**Funding:** MH received funding from the Wellcome Trust https://wellcome.org/ (Sir Henry Wellcome Postdoctoral Fellowship ref: 206470/Z/17/Z), British Heart Foundation https://www.bhf.org.uk/ (ref: PG/19/54/34511) and British Heart Foundation-Alan Turing Cardiovascular Data Science Award https://www.bhf.org.uk/for-professionals/information-for-researchers/what-we-fund/bhf-turing-cardiovascular-data-science-awards (ref: BHF-Turing-19/02/1022). JAB was funded by Wellcome Trust 4ward North Clinical Research Training Fellowship (ref: 227498/Z/23/Z). The funders had no role in study design, data collection and analysis, decision to publish, or preparation of the manuscript. The researchers have acted independently from funders and all authors had access to the data in the study and take responsibility for the integrity of the data and the accuracy of the data analysis.

**Competing interests:** I have read the journal's policy and the authors of this manuscript have the following competing interests: MH declares research grant income from the Wellcome Trust,

[7.7,7.9]), subsequent MI (7.1%; 95% CI [7.0,7.2]), and peripheral arterial disease (6.5%; 95% CI [6.4,6.6]). Compared with a risk-set matched population of 2,001,310 individuals, first hospitalisation of all non-fatal health outcomes were increased after MI, except for dementia (adjusted hazard ratio [aHR] 1.01; 95% CI [0.99,1.02]; $p = 0.468$) and cancer (aHR 0.56; 95% CI [0.56,0.57]; $p < 0.001$).

The study includes data from secondary care only—as such diagnoses made outside of secondary care may have been missed leading to the potential underestimation of the total burden of disease following MI.

## Conclusions

In this study, up to a third of patients with MI developed heart failure or renal failure, 7% had another MI, and 38% died within 9 years (compared with 35% deaths among matched individuals). The incidence of all health outcomes, except dementia and cancer, was higher than expected during the normal life course without MI following adjustment for age, sex, year, and socioeconomic deprivation. Efforts targeted to prevent or limit the accrual of chronic, multisystem disease states following MI are needed and should be guided by the demographic-specific risk charts derived in this study.

## Author summary

### Why was this study done?

- Myocardial infarction (MI; heart attack) can have major long-term impact on individuals and result in a wide range of further health conditions.

- Existing studies have focussed on determining the short-term risk of a second heart attack, stroke, or major bleeding, but research describing the long-term risk of major health outcomes for specific age, sex, and deprivation groups was lacking.

- Nationally representative and robust information of a wide range of long-term health outcomes following a heart attack is critical for the development of treatment recommendations, which take account of an individuals' specific risk.

### What did the researchers do and find?

- From the population of 56 million adults in England, we analysed hospital records for 34 million adults admitted to hospital (constituting 145 million admission records) to investigate the long-term health outcomes following a heart attack compared with individuals without a heart attack.

- Of 433,361 individuals with a heart attack, up to a third developed heart failure or renal failure, 7% had further heart attacks, and 38% died within the 9-year study period.

- Heart failure, atrial fibrillation, stroke, peripheral arterial disease, severe bleeding, renal failure, diabetes, and depression occurred more frequently for people who had a heart

British Heart Foundation and Alan Turing Institute. JAB declares research grant income from the Wellcome Trust. CPG has received funding, not in relation to this study, from Abbott Diabetes, Bristol Myers Squibb and the European Society of Cardiology, and consulting fees from AI Nexus, AstraZeneca, Amgen, Bayer, Bristol Myers Squibb, Boehrinher-Ingleheim, CardioMatics, Chiesi, Daiichi Sankyo, GPRI Research B.V., Menarini, Novartis, iRhyth, Organon as well as payment for honoraria or lectures from AstraZeneca, Boston Scientific, Menarini, Novartis, Raisio Group, Wondr Medical, Zydus. CPG declares participation on Data Safety Monitoring or Advisory boards for the DANBLCOK and TARGET CTCA trials and editorial and committee membership of the NICE Indicator Advisory Committee, EHJ Quality of Care and Clinical Outcomes and ESC Quality Indicator Committee. CH, LS, JW, HH, and PCL have no competing interests to declare.

**Abbreviations:** aHR, adjusted hazard ratio; CABG, coronary artery bypass graft; CI, confidence interval; CIF, cumulative incidence function; DAPT, dual antiplatelet therapy; GDPR, General Data Protection Regulation; GP, general practitioner; HDR UK, Health Data Research United Kingdom; HES, Hospital Episode Statistics; IMD, Index of Multiple Deprivation; LASER, Leeds Analytic Secure Environment for Research; MI, myocardial infarction; MINAP, Myocardial Ischaemia National Audit Project; NHS, National Health Service; PCI, percutaneous coronary intervention; PH, proportional hazard; PPIE, patient and public involvement and engagement; SD, standard deviation.

attack compared with those who did not, but the risk of cancer was lower overall and the risk of dementia did not differ overall.

## What do these findings mean?

- Efforts should be made to prevent or limit the development of long-term health outcomes that follow a heart attack—the likelihood of which differ depending on the age, sex, and deprivation of an individual.

- These findings are based on the full population of adults admitted to hospital in England, address limitations of previous studies, and can be used to inform preventative strategies tailored to specific individuals surviving a heart attack.

- The study was limited to hospitalisation data only—therefore, some diagnoses made outside of hospital may have been missed.

## Introduction

Information about the health outcomes of people with myocardial infarction (MI) is required to determine individual health needs, enable earlier detection and treatment of new onset disease, and inform health service planning. MI is a major contributor to further cardiovascular, renometabolic, and neuropsychiatric conditions [1–5]. Although estimating 10-year cardiovascular disease risk is an established part of primary prevention [6,7], comprehensive evidence for the long-term impact of MI on subsequent major health outcomes is lacking. Such information is critical not only for the development of guideline recommendations but also to underpin shared decision-making in the post-MI period [8]. Effective communication of the likely course of disease and risk of adverse long-term outcomes between individuals and healthcare professionals can promote positive lifestyle changes, facilitate treatment compliance, and improve patient understanding and quality of life [9,10].

Electronic health records are a powerful resource for understanding a diverse range of health outcomes over many years of follow-up [11]. While the largest study to date of post-MI health outcomes provides temporal trends over 2 decades (4.3 million patients in the United States (US)), outcomes were limited to 1-year mortality, readmission, and recurrent MI [4]. Indeed, the majority of studies of new onset disease following MI focussed only on short-term recurrent MI, bleeding, or stroke [12–26] (literature review S1 Text and S1 Table). Short- and long-term heart failure incidence following MI has been studied extensively [27–32]—but estimates vary widely (14% to 36%) [33]. While studies reporting the determinants of heart failure account for confounding and competing risk of death—absolute risk was commonly reported without adjustment, which is prone to bias and lacks generalisability [3,34–36]. Long-term post-MI incidence of atrial fibrillation [37–39] and depression [40–42] has been reported without adjustment for sociodemographic factors, pre-existing disease, and differential exposure times, and studies of depression were small (<300 patients). While data on the post-MI incidence of cancer (9% within 17 years; $n$ = 2.1 million) [43] and dementia (9% within 35 years, $n$ = 314,911) [2] were more robust—patient demographic-specific absolute risks remain unknown. To our knowledge, there were no contemporary, nationally representative studies of new onset peripheral arterial disease, chronic renal failure, or diabetes for survivors of MI (except one study of newly diagnosed diabetes at time of MI in young adults) [44].

To our knowledge, none of the studies reporting non-fatal health outcomes post-MI to date account for confounding as well as censoring and competing risks of death to quantify the absolute risk of outcomes over continuous follow-up time.

Understanding the clinical and public health importance of health outcomes after MI requires consideration of the absolute and relative risks beyond age- and sex-matched general populations. To the best of our knowledge, there are no studies of the long-term relative, absolute, and detailed patient demographic-specific risk of major cardiovascular and non-cardiovascular health outcomes following MI.

Therefore, we used hospital admission data in England to determine the risk of all-cause mortality and 11 non-fatal health outcomes following MI, including (1) health outcomes targeted through existing secondary prevention guidelines following MI (subsequent MI and heart failure); (2) health outcomes with shared risk factor profiles with MI, but which were not part of secondary prevention (peripheral arterial disease, cerebrovascular disease, and chronic renal failure); (3) health outcomes for which early detection is crucial for improved outcomes (severe bleeding, diabetes, atrial fibrillation, and cancer); and, finally, (4) health outcomes, which are difficult to prevent but have significant impact on individuals quality of life or life expectancy (depression and dementia). We hypothesised that post-MI disease incidence differed to that expected during a life course without MI. Therefore, we determined the excess incidence, adjusted absolute risk, and age, sex, deprivation, and time-specific risk for each of these health outcomes following MI compared with matched controls.

## Methods

We conducted a cohort study of all individuals aged $\geq$18 years admitted to one of 229 National Health Service (NHS) Trusts in England between 1 January 2008 and 31 January 2017. We analysed 11 non-fatal health outcomes identified a priori (subsequent MI and first hospitalisation for heart failure, atrial fibrillation, cerebrovascular disease, peripheral arterial disease, severe bleeding, renal failure, diabetes mellitus, dementia, depression, and cancer) and all-cause mortality for individuals hospitalised with MI compared with a risk-set matched control cohort.

Our study hypothesis was that risk of major health outcomes following MI differed from that expected during a life course without MI. The hypothesis and methodology were defined a priori—exceptions to this, including data-driven decisions and analyses conducted in response to peer review, are labelled as such throughout.

### Data access

Hospital Episode Statistics (HES) data were extracted from the Admitted Patient Care dataset by NHS Digital and linked with all-cause mortality data from the Office for National Statistics (censoring date 27 March 2017). HES data contain prospectively collected clinical, demographic, and organisational data for every hospitalisation to NHS hospitals in England, as previously described [45]. In brief, an individual's admission to hospital is recorded via a number of single episodes each containing a primary diagnosis, up to 19 secondary diagnoses and 24 operations, procedures, or interventions (coded according to the International Classification of Disease [ICD-10] and Office of Population Censuses and Surveys Classification of Interventions and Procedures [OPCS4.5], respectively) [45].

### Phenotype definitions

Individuals with MI and each of the 11 non-fatal health outcomes were identified using ICD-10 and OPCS4.5 codes derived from the Health Data Research United Kingdom (HDR UK)

Phenotype Library (healthdatagateway.org) (S2 Table) [46]. In addition, we identified ICD-10 codes for key subgroups including stroke, aortic disease, gastrointestinal bleeding, vascular dementia, acute and chronic renal failure, and colorectal, lung, breast, and prostate cancer (S2 Table).

Index MI, as well as the first occurrence of each outcome, was extracted from all primary and secondary diagnostic codes and all procedure codes across all hospitalisation episodes per individual. To identify index events within our study period, all patients with MI or any of the a priori health outcomes in any HES record prior to the study start date were excluded. We ascertained index MI from all diagnoses fields, as planned a priori, given that the presence of another dominant disease may impact on ascertainment from the first diagnostic position only [47]. Subsequent MI was defined as any MI more than 2 months from initial MI. We conducted sensitivity analyses for all outcomes in which follow-up was restricted to 2 or more months following MI to assess the impact of potential bias from the high number of events observed shortly after study entry. This was a data-driven decision (S2 Text). Data cleaning steps are outlined in S3 Text.

## Analytical cohort and matching process

Individuals were categorised into (1) a primary analytical cohort of those with an MI hospitalisation record and (2) a cohort of all hospitalised individuals who did not have MI (Fig 1). Our a priori analyses plans focussed on the comparison of outcomes between these 2 cohorts. However, in order to minimise bias arising from different demographic profiles between groups as well as to avoid immortal time bias, we instead employed an exact risk-set matching procedure [48]. This was determined prior to peer review, in favour of propensity score matching, to avoid the need for pruning of data resulting in efficiency loss and potential risk of reintroducing bias [49].

Risk-set matching involved matching any new case of MI occurring at time $t$ to any 5 individuals who had not yet developed MI by time $t$. Matching was based on single year of age, sex, month and year of hospital admission, and NHS Trust. Due to the longitudinal nature of our data, and as per risk-set matching guidance, individuals who later went on to develop MI were censored at the date of MI for the control cohort but contribute to both MI and matched control cohorts in analyses [48]. Study entry was either the date of the first episode with MI or first matched episode.

## Statistical analyses

Patient demographics including age (continuous), sex (male/female), year (single year of study entry), socioeconomic deprivation (Index of Multiple Deprivation [IMD] [50]—the official score of relative deprivation for small areas of England categorised into groups from least deprived [quintile 1] to most deprived [quintile 5]), crude mortality (Kaplan–Meier failure rate at 30 days, 1 year, and 5 years), total diagnoses by ICD-10 chapter heading, total person years of follow-up, and percentage of missing data for each demographic variable were summarised for the MI, non-MI, and matched control cohorts, respectively. Following peer review, baseline data relating to cardiovascular risk (including hypertension, dyslipidaemia, obesity, tobacco smoking, and alcohol excess) and use of an invasive coronary strategy for index MI (invasive coronary angiography, percutaneous coronary intervention [PCI], or coronary artery bypass graft [CABG]) (ICD-10 and OPCS4.5 code lists provided; S2 Table) were included. Baseline cardiovascular risk was based on diagnoses observed prior to, or on study entry, within the study period only as historical data were not available in-house due to information governance minimisation requirements (see Data governance section).

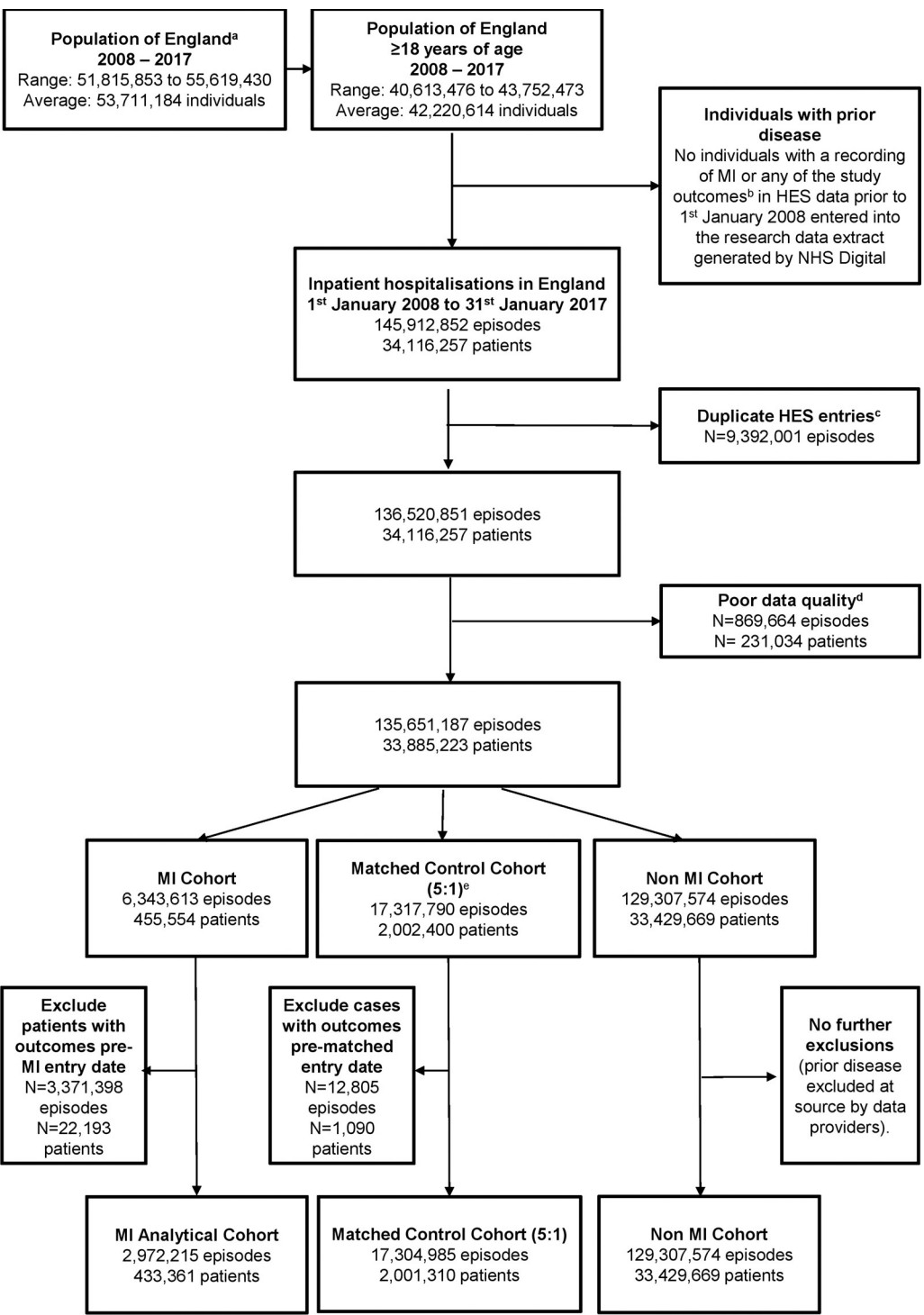

**Fig 1. Data extraction and cohort definitions for the population of England, 2008–2017.** [a]Population estimates extracted from the Office for National Statistics Population Estimates for England, Wales, Scotland, and Northern Ireland. [b]Heart failure, atrial fibrillation, cerebrovascular disease, peripheral vascular disease, severe bleeding, renal failure, diabetes mellitus, dementia, depression, and cancer. [c]Duplicate HES episodes, which contain the same data across admission start and end dates, episode start and end dates, primary and secondary diagnoses codes, and procedure codes, are administrative duplications where incorrect or new entries have been created and are removed from analyses (S3 Text). [d]Records that are missing core and essential information are deemed of too poor a quality to be included in analyses. These include records with missing, conflicting, or out-of-range finished consultant episode start and end dates, unknown spell begin and end indicators, or unknown episode order. [e]Individuals were matched according to single year

of age, sex, month and year of hospital admission, and NHS Trust using a 5:1 risk-set matching approach. HES, Hospital Episode Statistics; MI, myocardial infarction; NHS, National Health Service.

**Excess rate of health outcomes and all-cause mortality.** Unadjusted rates of disease per 1,000 person-years of follow-up, attained age, and adjusted excess rate of post-MI disease for each outcome compared with matched controls were calculated. Excess rate of post-MI disease was based on adjusted hazard ratios (aHRs) and 95% confidence intervals (CIs) from flexible parametric survival models per outcome [51]. Models adjusted for age, sex, calendar year, and deprivation score. Nonlinearity of age was modelled using restricted cubic splines (3 degrees of freedom). To provide an overall estimate of excess risk of post-MI disease, comparable with existing literature, aHRs were modelled as standard proportional hazards [PHs] (i.e., fixed constant over time). Subsequently, the PH assumption was relaxed to provide higher resolution insight into absolute risk of outcomes over continuous follow-up time (described below).

**Absolute risk of health outcomes and all-cause mortality.** The adjusted absolute risk of each outcome was calculated through standardised cumulative incidence functions (CIFs) stratified by a cohort over 9 years of follow-up, based on the same set of adjusted survival models specified above, treating death without outcome as a competing risk and additionally including a time-dependent effect for MI versus matched controls to relax the PH assumption and reflect variation in the difference in cumulative incidence between cohorts over continuous follow-up time (using "standsurv" in Stata MP v17) [52].

**Age, sex, and deprivation-specific risk charts.** Standardised cumulative incidence for each outcome were calculated for all combinations of age group (<40, 40 to <50, 50 to <60, 60 to <70, 70 to <80, 80 to <90, and ≥90 years of age), sex, and socioeconomic deprivation quintile. Following peer review, age, sex, and deprivation-specific risk for the MI cohort were additionally adjusted for receipt of invasive coronary angiography, PCI, or CABG at time of MI. Risk charts were generated for each cohort at 60 days, 1 year, and 5 years of follow-up and presented in heat maps and an interactive web-based application.

Multiple imputation for missing data was not performed owing to (i) the minimal amount of missing data in core data fields and (ii) the significant increase in computational power required versus the minimal gain in analytical accuracy for data of this scale.

## Data governance, IT infrastructure, ethics, and reporting standards

This research adheres to General Data Protection Regulation (GDPR) (privacy notice). Data minimisation standards were met through pseudonymisation, month/year aggregation of date, exclusion of patients aged <18 years old, and individuals with pre-existing conditions from a priori outcomes at source. Data were stored and processed within the Leeds Analytic Secure Environment for Research (LASER), University of Leeds. Ethical approval was not required for this study, which solely relies on the secondary use of routinely collected, non-confidential healthcare data. This study is reported as per the Reporting of studies Conducted using Observational Routinely-collected Data (RECORD) guideline and CODE-EHR minimum standards (S1 and S2 Checklists).

## Patient and public involvement and engagement (PPIE)

In the period prior to obtaining research funding, the research team consulted with individuals who have had, or cared for someone who had, an MI. Shared experiences of those individuals shaped early study design and reporting of this research. Individuals raised concerns about the lack of information provided regarding future health risks following their heart attack. They

noted that there was a particular focus on changes in diet and exercise in the immediate period after a heart attack, with little or no information available of "red flags to look out for" in the long term. Individuals identified the need for tools to enable doctors to "risk assess us" and tell us more about future health prospects. These discussions directly informed our research design and ensured we retained a long-term focus, instead of curtailing outcomes at 1-year post-MI to align with existing evidence. Furthermore, we developed an interactive, open-access tool that can be used by healthcare professionals, individual patients, and their carers to better understand and communicate the absolute risks of developing a range of health outcomes motivated by our patient and public involvement and engagement (PPIE) discussions. Individuals felt that this knowledge may provide greater incentivisation for positive lifestyle changes following a heart attack as well as allow individuals and healthcare professionals to "act early rather than react late."

Finally, the research team hosted a workshop attended by a further 8 individuals with cardiovascular and other multiple long-term health conditions providing direct feedback on this study and guiding the dissemination strategy and direction of follow-on studies. The PPIE group identified the need for joined up thinking between different healthcare providers, given the risk of both cardiovascular and non-cardiovascular conditions, which we raise in our manuscript discussion. They noted the adoption of a long-term perspective is particularly important given that many expect good life expectancy after MI. The PPIE group advocated for the importance of dissemination to general practitioners (GPs) as well as cardiologists and identified the need for clear lay summaries of the work to ensure it is accessible by all. The research team will coproduce lay summaries with our PPIE members and disseminate findings to relevant patient groups, including through the British Heart Foundation's Heart Voices network.

## Results

Of the 55,619,430 populace of England, 34,116,257 individuals aged 18 years and above were admitted to hospital amounting to 145,912,852 hospital episodes in NHS hospitals in England over the study period. The analytical cohorts comprised 433,361 individuals with MI (2,972,215 episodes), 33,429,669 individuals without MI (129,307,574 episodes), and a subset of 2,001,310 matched controls (17,304,985 episodes) (Fig 1). There were 18,322 matched controls who went on to develop MI (0.92%) and therefore contribute data to both cohorts. Individuals with MI were admitted to hospital at a mean age of 67.4 years (standard deviation [SD] 14.4), were predominantly male (65.5% [$n = 283,742$]), and had a 30-day mortality rate of 9.9% [$n = 42,882$] (Table 1). For matched controls, the age and sex profile was similar to those with MI by design (mean age 66.8 [SD 14.2] and [65.7% male, $n = 1,314,388$]) with 3.1% ($n = 59,991$) 30-day mortality. There were minimal differences in deprivation between cohorts (20.7% [$n = 77,008$] versus 19.2% [$n = 375,734$] were in the most deprived category for MI and matched controls, respectively). The proportion of individuals with hypertension, dyslipidaemia, obesity, and tobacco smoking were higher among those with MI compared with matched controls but lower for alcohol excess (Table 1). A total of 319,439 (73.7%) individuals with MI received an invasive coronary management strategy for index MI. Missing data in core data fields were limited, including 50,438 (0.1%) for age, 33,871 (0.1%) for sex, 0 missing for month and year of admission, but higher for deprivation ($n = 4,025,757$, 11.8%).

There were 18,343,361 diagnoses codes covering all conditions for individuals with MI throughout the study period, of which 24.7% [$n = 4,526,829$] were unique nonfatal diagnoses codes per individual (Table 1). The majority of diagnosis codes related to the circulatory system (44.3% [$n = 2,003,429$]), which, along with "endocrine, nutritional, and metabolic diseases" (9.6% [$n = 433,545$]), appeared more frequently than in matched controls (20.2% [$n = 3,631,165$] and 9.0% [$n = 1,616,515$] for circulatory and "endocrine, nutritional, and metabolic diseases,"

**Table 1. Patient characteristics for people with MI, without MI, and age, sex, and year matched controls hospitalised in England, 2008–2017.**

| | Analytical Cohort | | |
|---|---|---|---|
| | **Non-MI** $N = 33,429,669$ | **MI** $N = 433,361$ | **Matched control** $N = 2,001,310$ |
| **Age at study entry, years (mean, SD)[a]** | 41.7 (26.1) | 67.4 (14.4) | 66.8 (14.2) |
| **Male Sex (Numerator/denominator, %)** | 14,747,198/33,395,663 (44.2%) | 283,742/443,298 (65.5%) | 1,314,388 (65.7%) |
| **Study entry period (Numerator/denominator, %)** | | | |
| 2008–2010 | 14,473,148/33,429,669 (43.3%) | 155,563/433,361 (35.7%) | 700,484/2,001,310 (35.0%) |
| 2011–2013 | 10,428,354/33,429,669 (31.2%) | 143,841/433,361 (33.3%) | 704,323/2,001,310 (35.2%) |
| 2014–2017 | 8,528,167/33,429,669 (25.5%) | 133,957/433,361 (31.1%) | 596,503/2,001,310 (29.8%) |
| **Socioeconomic Status (Numerator/denominator, %)[b]** | | | |
| 1 –Least deprived | 5,474,307/29,417,676 (18.6%) | 77,008/419,597 (18.4%) | 383,572/1,954,523 (19.6%) |
| 2 | 5,684,181/29,417,676 (19.3%) | 84,328/419,597 (20.1%) | 410,442/1,954,523 (21.0%) |
| 3 | 5,883,886/29,417,676 (20.0%) | 86,689/419,597 (20.7%) | 405,714/1,954,523 (20.8%) |
| 4 | 6,061,653/29,417,676 (20.6%) | 84,751/419,597 (20.2%) | 379,061/1,954,523 (19.4%) |
| 5 –Most deprived | 6,313,649/29,417,676 (21.5%) | 86,821/419,597 (20.7%) | 375,734/1,954,523 (19.2%) |
| **Cardiovascular risk factors (N, %)[c]** | | | |
| Hypertension | 6,839,977/33,429,669 (20.5%) | 211,386/433,361 (48.8%) | 738,075/2,001,310 (36.9%) |
| Dyslipidaemia | 2,360,150/33,429,669 (7.1%) | 127,921/433,361 (29.5%) | 237,305/2,001,310 (11.9%) |
| Obesity | 1,541,944/33,429,669 (4.6%) | 25,706/433,361 (5.9%) | 74,851/2,001,310 (3.7%) |
| Tobacco smoking | 3,974,449/33,429,669 (11.9%) | 124,773/433,361 (28.8%) | 228,557/2,001,310 (11.4%) |
| Alcohol excess | 1,316,047/33,429,669 (3.9%) | 17,031/433,361 (3.9%) | 92,905/2,001,310 (4.6%) |
| **Invasive coronary strategy[d]** | | | |
| Invasive coronary angiography, PCI, or CABG | NA | 319,439/433,361 (73.7%) | NA |
| **Crude mortality (N, KM failure function)** | | | |
| 30 days | 357,416 (1.1%) | 42,882 (9.9%) | 59,991 (3.1%) |
| 1 year | 1,030,354 (3.1%) | 67,356 (15.7%) | 202,713 (10.3%) |
| 5 years | 2,416,967 (8.6%) | 106,958 (28.6%) | 435,196 (26.1%) |
| Total follow-up period (9 years) | 3,003,970 (13.9%) | 119,695 (39.6%) | 496,180 (37.3%) |
| **Total primary and secondary diagnoses (ICD-10 Chapters I–XIV)[e] (Numerator/denominator, %)** | | | |
| Certain infectious and parasitic diseases (A00–B99) | 4,207,444/134,983,929 (3.1%) | 105,005/4,526,829 (2.3%) | 632,462/17,995,217 (3.5%) |
| Neoplasms (C00–D48) | 4,861,221/134,983,929 (3.6%) | 91,264/4,526,829 (2.0%) | 1,064,275/17,995,217 (5.9%) |
| Diseases of the blood and blood-forming organs (D50–D89) | 2,672,524/134,983,929 (2.0%) | 91,216/4,526,829 (2.0%) | 489,230/17,995,217 (2.7%) |
| Endocrine, nutritional, and metabolic diseases (E00-E90) | 7,472,910/134,983,929 (5.5%) | 433,545/4,526,829 (9.6%) | 1,616,515/17,995,217 (9.0%) |

(*Continued*)

**Table 1.** (Continued)

| | Analytical Cohort | | |
|---|---|---|---|
| | **Non-MI** $N = 33,429,669$ | **MI** $N = 433,361$ | **Matched control** $N = 2,001,310$ |
| Mental and behavioural disorders (F00-F99) | 6,575,834/134,983,929 (4.9%) | 224,846/4,526,829 (5.0%) | 1,001,335/17,995,217 (5.6%) |
| Diseases of the nervous system (G00–G99) | 3,400,963/134,983,929 (2.5%) | 77,527/4,526,829 (1.7%) | 553,259/17,995,217 (3.1%) |
| Diseases of the eye and adnexa (H00–H59) | 3,311,967/134,983,929 (2.5%) | 108,405/4,526,829 (2.4%) | 755,446/17,995,217 (4.2%) |
| Diseases of the ear and mastoid process (H60–H95) | 971,522/134,983,929 (0.7%) | 23,627/4,526,829 (0.5%) | 141,062/17,995,217 (0.8%) |
| Diseases of the circulatory system (I00–I99) | 9,899,036/134,983,929 (7.3%) | 2,003,429/4,526,829 (44.3%) | 3,631,165/17,995,217 (20.2%) |
| Diseases of the respiratory system (J00–J99) | 7,663,859/134,983,929 (5.7%) | 348,782/4,526,829 (7.7%) | 1,569,536/17,995,217 (8.7%) |
| Diseases of the digestive system (K00–K93) | 10,866,943/134,983,929 (8.1%) | 396,322/4,526,829 (8.8%) | 2,646,415/17,995,217 (14.7%) |
| Diseases of the skin and subcutaneous tissue (L00–L99) | 3,193,502/134,983,929 (2.4%) | 76,490/4,526,829 (1.7%) | 507,402/17,995,217 (2.8%) |
| Diseases of the musculoskeletal system and connective tissue (M00–M99) | 7,844,158/134,983,929 (5.8%) | 296,028/4,526,829 (6.5%) | 1,905,382/17,995,217 (10.6%) |
| Diseases of the genitourinary system (N00–N99) | 7,159,761/134,983,929 (5.3%) | 250,343/4,526,829 (5.5%) | 1,481,733/17,995,217 (8.2%) |
| **Total unique 3-digit ICD10 codes[f]** | 134,983,929 | 4,526,829 | 17,995,217 |
| **Total non-unique 3-digit ICD10 codes[g]** | 340,047,260 | 18,343,361 | 84,154,930 |
| **Total person-years of follow-up** | 165,450,465 | 1,603,181 | 7,902,483 |
| **Total episodes, N** | **129,307,574** | **2,972,215** | **17,304,985** |

[a]Age at study entry reflects age at first MI, age at first hospitalisation for any cause, and age at first matched hospitalisation for any cause for the MI, non-MI, and matched control cohorts, respectively.

[b]Measured according to quintiles of the IMD.

[c]Baseline cardiovascular risk data relate to diagnoses codes recorded on or prior to individual study entry dates but do not include data prior to 1 January 2008. In addition, tobacco smoking, alcohol excess, and obesity are known to be under recorded in hospitalisation data and likely represent only the extremes of the patient population.

[d]Invasive coronary angiography, PCI, or CABG.

[e]Each unique 3-digit ICD-10 code is counted only once per individual before aggregating to ICD-10 chapter heading level providing a summary of diagnoses codes across the study period.

[f]Any 3-digit ICD10 diagnoses code, counted once per individual, within ICD-10 chapters I to XIV (A00 to N99), and serves as the denominator for chapter heading counts.

[g]Any 3-digit diagnoses code in ICD-10 chapters 1 to 14 (A00-N99), including repeated diagnoses per individual. Missing data were minimal including 50,438 (0.1%) for age, 33,871 (0.1%) for sex, 0 missing for month and year of admission, and 4,025,757 (11.8%) for socioeconomic deprivation across the analytical cohorts.

CABG, coronary artery bypass graft; ICD, International Classification of Diseases; KM, Kaplan–Meier; MI, myocardial infarction; NA, not applicable; PCI, percutaneous coronary intervention; SD, standard deviation.

respectively). For matched controls, diagnoses relating to neoplasms (20.2%, [$n = 3,631,165$]) and diseases of the digestive system (14.7% [$n = 2,646,415$]) occurred most frequently (Table 1).

## Excess rate of health outcomes and all-cause mortality

The most frequent health outcomes following MI, prior to standardisation, were heart failure (crude rate 86.4; 95% CI [85.9,86.9] per 1,000 person-years [1,000 pyrs]), atrial fibrillation (64.3; 95% CI [63.9,64.7] per 1,000 pyrs), renal failure (56.5; 95% CI [56.1,56.9] per 1,000 pyrs),

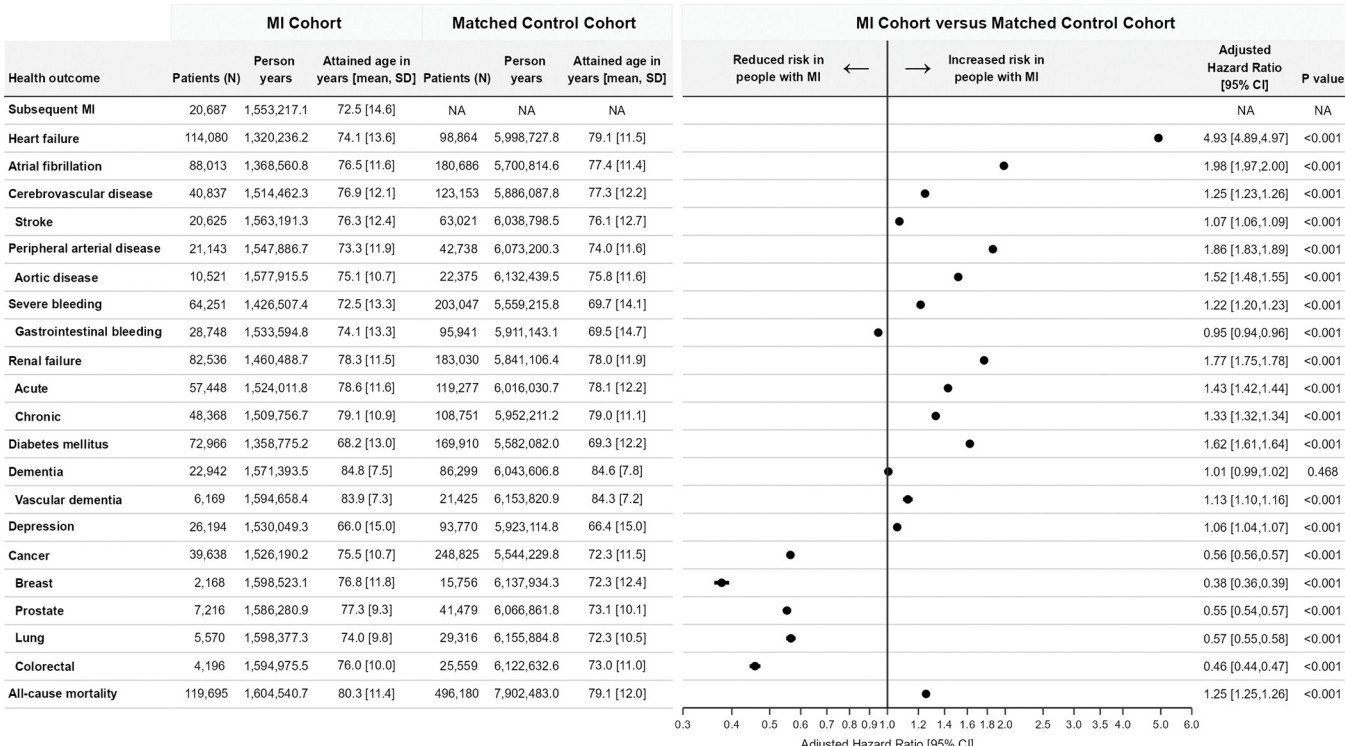

**Fig 2. Number of individuals, attained age, and excess rate[a] of 11 nonfatal health outcomes, 9 key subgroups, and all-cause mortality following MI compared with age, sex, and year matched controls in England.** [a]Excess rate of post-MI hospitalisations and all-cause mortality presented as aHRs and 95% CIs based on a series of flexible parametric survival models for each outcome adjusted for age at admission, sex, calendar year of admission, and deprivation score. Age was modelled using restricted cubic spline functions with 3 degrees of freedom to allow for its potential nonlinear association with outcomes, and death without event was treated as a competing risk. Complimentary sensitivity analyses, in which follow up was restricted to begin a minimum of 2 months after study entry, are provided in S8 Table. aHR, adjusted hazard ratio; CI, confidence interval; MI, myocardial infarction; NA, not applicable; SD, standard deviation.

and diabetes mellitus (53.7; 95% CI [53.3,54.1] per 1,000 pyrs) (S3 and S4 Tables). There was an excess rate of heart failure (aHR 4.93; 95% CI [4.89,7.97]; $p < 0.001$), atrial fibrillation (aHR 1.98; 95% CI [1.97,2.00]; $p < 0.001$), cerebrovascular disease (aHR 1.25; 95% CI [1.23,1.26]; $p < 0.001$), peripheral arterial disease (aHR 1.86; 95% CI [1.83,1.89]; $p < 0.001$), severe bleeding (aHR 1.22; 95% CI [1.20,1.23]; $p < 0.001$), renal failure (aHR 1.77; 95% CI [1.75,1.78]; $p < 0.001$), diabetes mellitus (aHR 1.62; 95% CI [1.61,1.64]; $p < 0.001$), vascular dementia (aHR 1.13; 95% CI [1.10,1.16]; $p < 0.001$), and depression (aHR 1.06; 95% CI [1.04,1.07]; $p < 0.001$) following MI compared with matched controls (Fig 2). There was no difference in the rate of dementia overall (aHR 1.01; 95% CI [0.99,1.02]; $p = 0.468$) and a reduced rate of cancer (aHR 0.56; 95% CI [0.56,0.57]; $p < 0.001$) (Fig 2).

## Absolute risk health outcomes and all-cause mortality

Overall, the adjusted cumulative incidence at 9 years post-MI was highest for all-cause mortality (37.8%; 95% CI [37.6,37.9]) followed by heart failure (29.6%; 95% CI [29.4,29.7]), renal failure (27.2%; 95% CI [27.0,27.4]), atrial fibrillation (22.3%; 95% CI [22.2,22.5]), severe bleeding (19.0%; 95% CI [18.8,19.1]), diabetes mellitus (17.0%; 95% CI [16.9,17.1]), cancer (13.5%, 95% CI [13.3,13.6]), cerebrovascular disease (12.5%; 95% CI [12.4,12.7]), depression (8.9%; 95% CI [8.7,9.0]), dementia (7.8%; 95% CI [7.7,7.9]), subsequent MI (7.1%; 95% CI [7.0─7.2]), and peripheral arterial disease (6.5%; 95% CI [6.3,6.5]) (Figs 3 and 4 and S5 Table). Cumulative

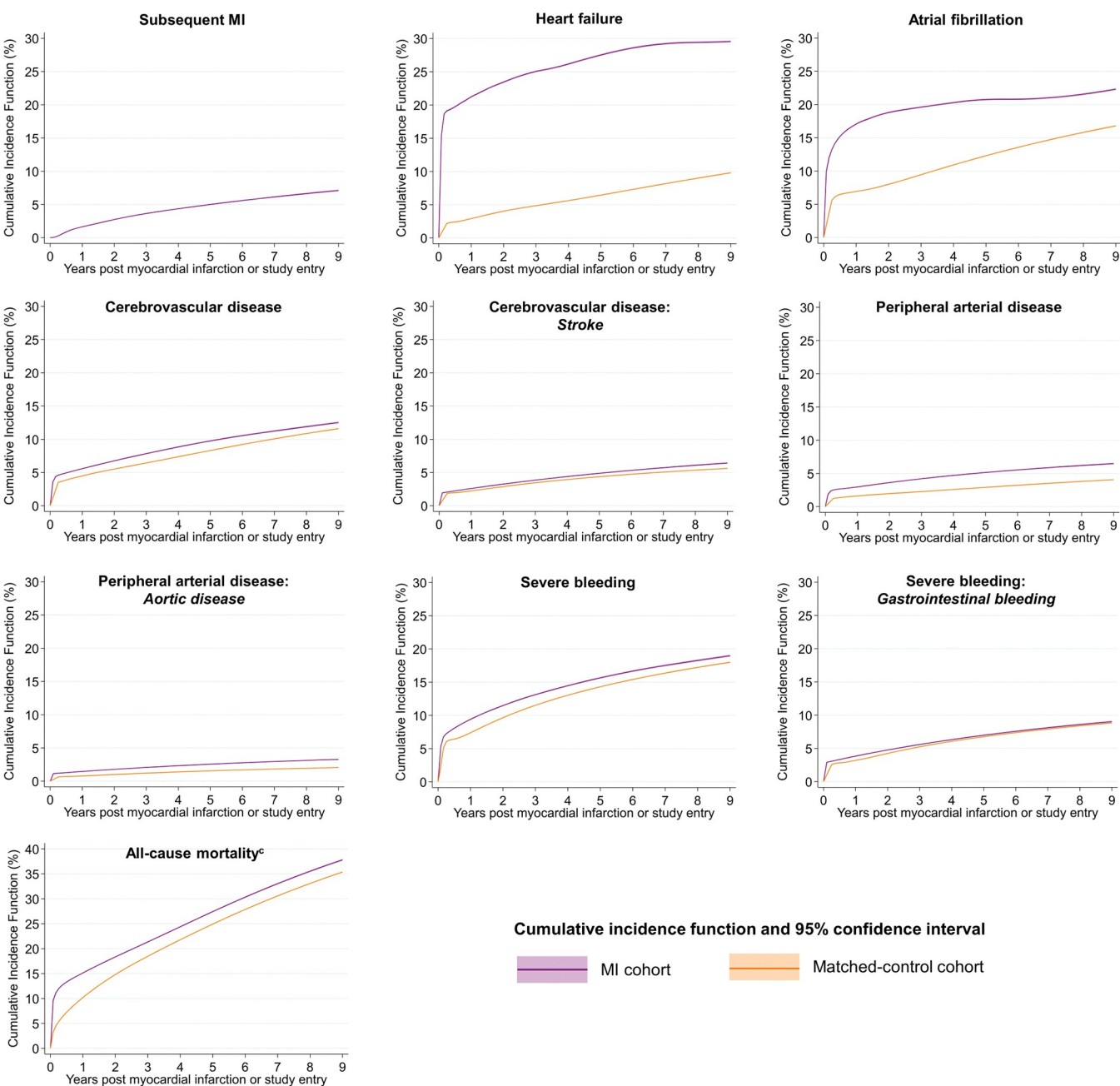

**Fig 3. Adjusted absolute risk[a] over continuous time of all-cause mortality, subsequent MI, heart failure, atrial fibrillation, cerebrovascular disease, peripheral arterial disease, and severe bleeding following MI compared with matched controls[b] in England.** [a]Calculated according to the standardised CIF, treating death without outcome as a competing risk, adjusted for nonlinear age using restricted cubic splines, sex, calendar year and deprivation score and a time-dependent effect for MI versus matched controls. Full CIFs and CIs by time point provided in S5 Table, and sensitivity analyses, in which follow-up was restricted to begin a minimum of 2 months after study entry, presented in S1 Fig and S6 Table. Numbers at risk at 1, 5, and 9 years of follow-up are provided in S7 Table. [b]Individuals were matched according to single year of age, sex, month and year of hospital admission, and NHS Trust using a 5:1 risk-set matching approach. [c]y-Axis range for all-cause mortality differs to plots for nonfatal health outcomes. CI, confidence interval; CIF, cumulative incidence function; ICD, International Classification of Disease; MI, myocardial infarction; NHS, National Health Service.

incidence was greater among the MI cohort compared with matched controls for all outcomes except gastrointestinal bleeding—where it was higher in the short term (3.8%; 95% CI [3.8,3.9] versus 3.2% 95% CI [3.1,3.2] at 1 year) and similar in the long term (9.0%; 95% CI [8.9,9.1]

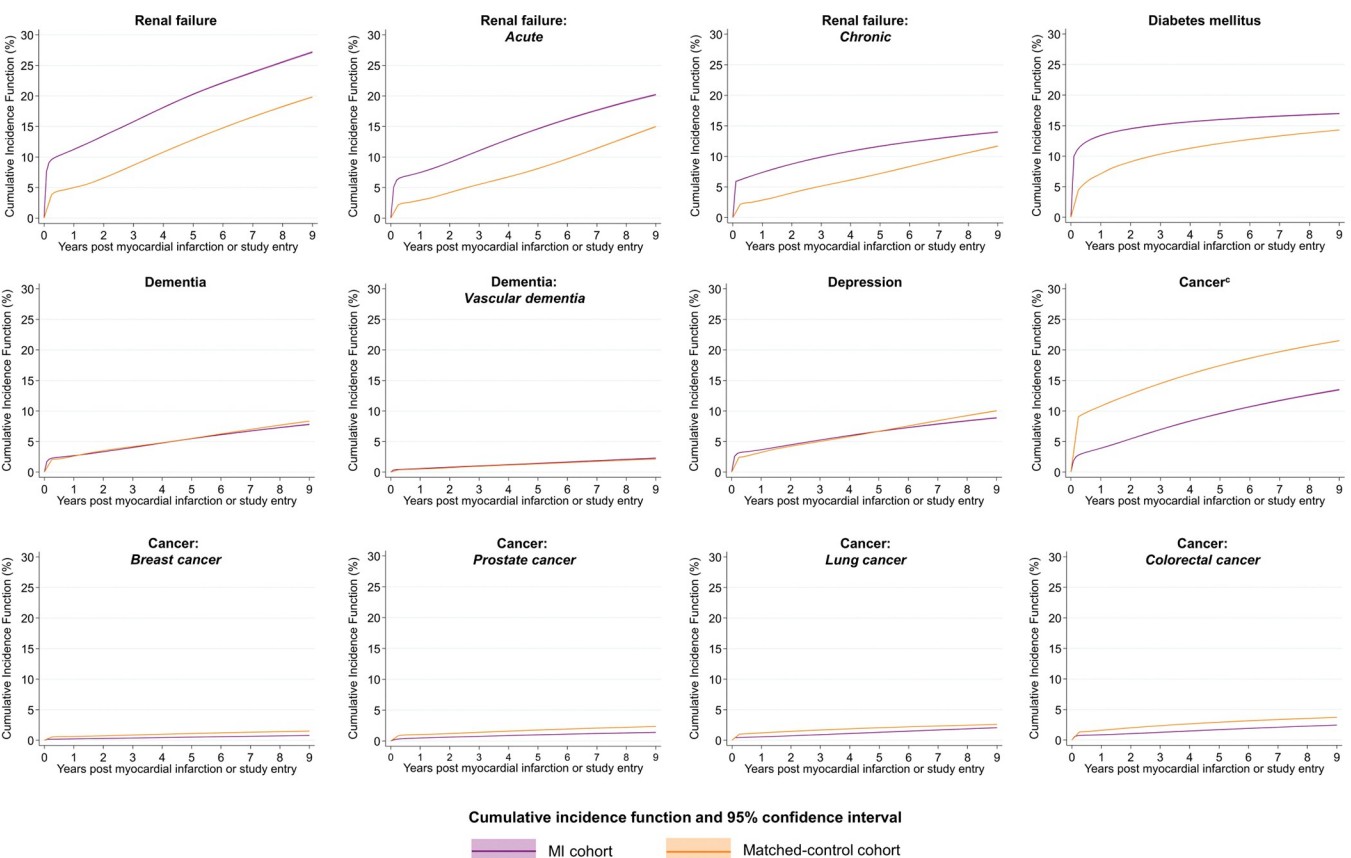

**Fig 4. Adjusted absolute risk[a] over continuous time of renal failure, diabetes mellitus, dementia, depression, and cancer following MI compared with matched controls[b] in England.** [a]Calculated according to the standardised CIF, treating death without outcome as a competing risk, adjusted for nonlinear age using restricted cubic splines, sex, calendar year and deprivation score and a time-dependent effect for MI versus matched controls. Full CIFs and CIs by time point provided in S5 Table, and sensitivity analyses, in which follow-up was restricted to begin a minimum of 2 months after study entry, presented in S2 Fig and S6 Table. Numbers at risk at 1, 5, and 9 years of follow-up are provided in S7 Table. [b]Individuals were matched according to single year of age, sex, month and year of hospital admission, and NHS Trust using a 5:1 risk-set matching approach. [c]Includes all cancer types (ICD10 codes C00–C97), i.e., this category is not restricted to the sum of breast, prostate, lung, and colorectal cancer). CI, confidence interval; CIF, cumulative incidence function; ICD, International Classification of Disease; MI, myocardial infarction; NHS, National Health Service.

versus 8.8%; 95% CI [8.8,8.9]); dementia—where incidence was higher in the short term (2.1%; 95% CI [2.1,2.2] versus 1.79; 95% CI [1.77,1.81] at 60 days) and lower in the long term (7.8%; 95% CI [7.7,7.9] vesus 8.34%; 95% CI [8.28,8.41] at 9 years); and cancer—where incidence was lower throughout the follow-up period (13.5%; 95% CI [13.3,13.6] versus 21.5%; 95% CI [21.4,21.6] at 9 years) (sensitivity analyses S1 and S2 Figs and S6 Table; numbers at risk S7 Table).

## Age, sex, and deprivation-specific risk charts for 11 nonfatal health outcomes and all-cause mortality following MI

There was an increasing risk with age post-MI for men and women across all deprivation quintiles for heart failure (cumulative incidence at 5 years ranging from 13.5%; 95% CI [13.0,14.0] to 48.9%; 95% CI [48.0,49.7] for men in deprivation quintile 3 aged <40 years and ≥90 years, respectively), atrial fibrillation (2.5%; 95% CI [2.3,2.7] to 36.6%; 95% CI [35.8,37.4] for men in deprivation quintile 3 aged <40 years and ≥90 years, respectively), and renal failure (4.0%; 95% CI [3.7,4.3] to 46.8%; 95% CI [45.9,47.7]) for men in deprivation quintile 3 aged <40

years and ≥90 years, respectively) (Fig 5). The association of age with subsequent MI, peripheral arterial disease, cerebrovascular disease, diabetes, and cancer was less pronounced. In contrast, post-MI depression risk was highest among younger age groups at each time point, particularly for those in the most deprived quintile (5-year CIFs for men in deprivation quintile 5 were 11.5%; 95% CI [10.9,12.0]; 10.0%; 95% CI [9.7,10.3]; 8.5%; 95% CI [8.2,8.7]; 6.8%; 95% CI [6.6,6.9]; 5.3%; 95% CI [5.1,5.5]; 4.4%; 95% CI [4.3,3.6]; and 3.8%; 95% CI [3.5,4.0] for those aged <40 years, 40 to <50 years, 50 to <60 years, 60 <70 years, 70 to <80 years, 80 to <90 years, and ≥90 years respectively). The risk of depression post-MI was also higher among women compared with men (5-year CIFs for women 21.5%; 95% CI [20.5,22.5], 18.9%; 95% CI [18.3,19.5]; 16.1%; 95% CI [15.6,16.6]; 13.0%; 95% CI [12.6,13.4]; 10.3%; 95% CI [10.0,10.6]; 8.7%; 95% CI [8.4,9.0]; 7.5%; 95% CI [7.0,7.9] for those aged <40 years, 40 to <50 years, 50 to <60 years, 60 to <70 years, 70 to <80 years, 80 to<90 years, and ≥90 years, respectively (Fig 5). Complimentary risk charts for matched controls and sensitivity analyses provided in S3, S4 and S5 Figs and interactive versions accessed via https://multimorbidity-research-leeds.github.io/research-resources.

## Discussion

In this study of over 145 million hospitalisations in England, we provide nationwide evidence from a single health system of the specific burden of a wide range of health outcomes following MI. Up to a third of patients with MI developed heart failure or renal failure, 13% cerebrovascular disease, 9% depression, 7% had further MI or peripheral arterial disease, and 38% died within 9 years (compared with 35% deaths for individuals without MI). Rates of all health outcomes, except dementia and cancer, were significantly higher than expected during the normal life course without MI.

Increased incidence of heart failure post-MI is well recognised [3], but estimates have been inconsistent, often lack confounder adjustment, and were unavailable by detailed demographic groups. Our study provides adjusted estimates of 21.2% heart failure at 1 year, rising to 29.6% at 9 years following MI compared with 2.9% and 9.8% at 1 and 9 years for matched controls, respectively. Further, our study shows earlier onset of heart failure following MI for the most socioeconomically deprived individuals. While we did not assess secondary preventative medication directly, our findings may reflect previously reported underuse of secondary preventative medication among socioeconomically deprived groups after MI [53]. We found that almost one-fifth of patients were admitted to hospital with severe bleeding following MI. While we were unable to study dual antiplatelet therapy (DAPT), recent data indicated that a reduction of major bleeding complications may be achieved through shortened DAPT regimes [54]. The incidence of post-MI diabetes mellitus, peripheral vascular disease, and renal failure had not been previously reported. Here, we quantify a small excess incidence of diabetes mellitus (17% versus 14%) and peripheral arterial disease (6% versus 4%) at 9 years, and a marked difference in the incidence of renal failure (27% versus 20%) following MI compared with controls.

New hospitalisations for depression occurred in 1 in 11 individuals after MI and was more frequent at younger ages of MI, for those in the most deprived quintile, and among women. Given the increasing trend in cardiovascular risk factors among young adults [55], and the increasing proportion of MI among young adults and women [56], the incidence of depression following MI will likely rise.

Our study showed a lower incidence of cancer overall following MI, as well as for breast, colorectal, lung, and prostate cancer compared with controls. Two existing large-scale studies also describe significantly reduced risks of breast and prostate cancer among individuals with

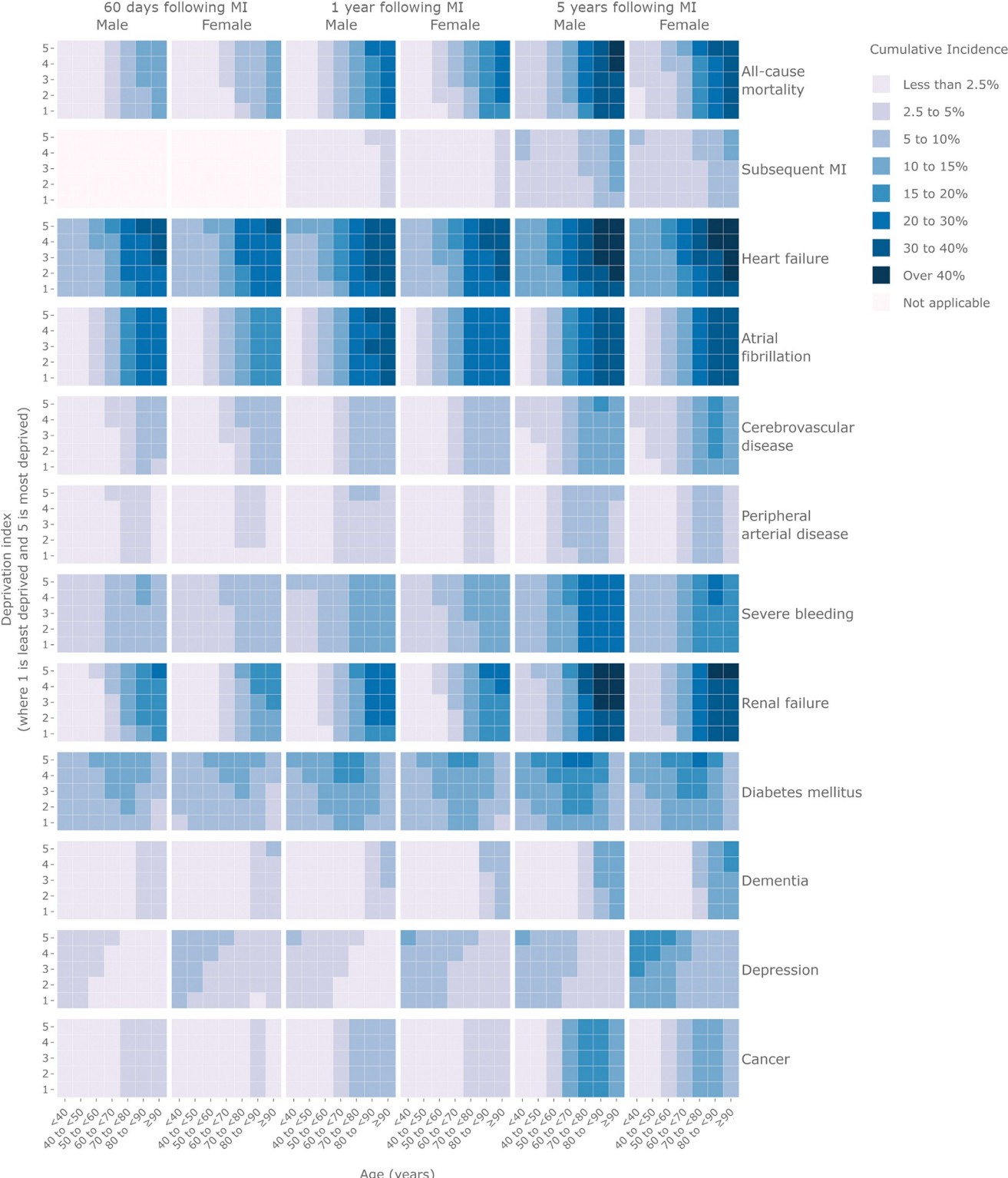

**Fig 5. Absolute risk[a] of subsequent MI, heart failure, atrial fibrillation, cerebrovascular disease, peripheral arterial disease, severe bleeding, renal failure, diabetes mellitus, dementia, depression, cancer, and all-cause mortality at 60 days, 1 year, and 5 years following MI by age group, sex, and deprivation[b] in England (*N* = 433,361 individuals).** [a]Calculated according to the standardised CIF, treating death without outcome as a competing risk and adjusted for nonlinear age using restricted cubic spline functions, sex, calendar year, deprivation score, and receipt of invasive coronary angiography, percutaneous coronary intervention, or coronary artery bypass graft. [b]Deprivation is measured using the IMD where 1 indicates those in the least deprived

fifth, and 5 indicates those in the most deprived fifth. S3, S4 and S5 Figs show CIFs for the matched control cohort and the main and matched control sensitivity analyses. Interactive version of these data are provided (https://multimorbidity-research-leeds.github.io/research-resources). CIF, cumulative incidence function; IMD, Index of Multiple Deprivation; MI, myocardial infarction.

MI [43] and cardiovascular disease more broadly [57]. Our study builds on this, providing demographic-specific absolute risk overall and for key cancer subgroups. In contrast with previously reported data, we additionally show a reduced risk of lung and colorectal cancer and all cancers combined (compared with previously reported increased incidence of lung cancer and nonsignificant difference in colorectal cancer following MI [43] and increased risk of lung, colorectal, and all cancers for individuals with cardiovascular disease [57]). The reason for conflicting evidence may be explained by (1) different population demographics of previously published work (smaller and younger population of MI [43] and broader cardiovascular population [57]) and (2) lack of accounting for competing risk of death [57], which was likely to have a marked impact on findings given the high early mortality rate observed for individuals with cardiovascular disease. Mechanisms underpinning reduced risk of cancer following MI remain unclear and warrant further investigation. While trial data have assessed the role of aspirin in the onset, progression, and mortality of specific cancer subtypes [58–60], broader evidence for its beneficial effect remains unclear [61]. We are unable to state whether aspirin contributed to the reduced risk of cancer observed, given medication data were unavailable. While some evidence points to reduced screening for cancer among individuals with cardiovascular disease [62], a surveillance bias may also act in the opposite direction. Finally, given that we focussed specifically on development of new cancer following MI, we included only individuals living long enough cancer-free to develop MI before cancer, and this may further explain our findings.

The incidence of dementia overall was 7.8% and 2.3% for vascular dementia within 9 years of MI. There was no difference in the risk of dementia overall, but there was a 13% increased risk of vascular dementia following MI compared with controls—consistent with previous work [2]. We improve on previous data by showing an increased risk of dementia overall in the short term (2.1% versus 1.8% at 60 days) but reduced risk in the long term (7.8% versus 8.3% at 9 years) and a consistently increased risk of vascular dementia at all time points following MI compared with controls.

While we use large-scale nationwide data and robust methodology to produce generalisable results, we do acknowledge the study limitations. Our focus was on hospitalised events and we did not have access to diagnoses made in primary care–with may have (1) underestimated the totality of post-MI disease and (2) led to some individuals with pre-existing comorbidities being missed from our exclusion criteria—which may vary by outcome. While the steep increase in events after study entry could in part reflect underestimation of pre-existing disease, the observed pattern of events was expected given that individuals enter the study at a key clinical event, signifying more severe or complex disease, rather than in a healthy state. To further mitigate this risk, we present sensitivity analyses delaying follow up to 60 days after study entry, while acknowledging that our primary findings are substantiated by other studies reporting high rates of rehospitalisation within 30 days following MI [63], reflecting conditions likely diagnosed shortly after MI due to screening for risk factors (e.g., diabetes) and sequelae (e.g., heart failure) or due to complications (e.g., bleeding). We further ensured high ascertainment of preexisting disease by making use of a look-back period in excess of 10 years and capturing conditions via comprehensive code lists. We acknowledge that, for some individuals, the look-back period may have been limited (e.g., due to immigration), which we could not quantify.

Case ascertainment of MI within HES is known to be high; indeed, individuals with long-term conditions as well as MI are more likely to be captured by HES than by the UK's Myocardial Ischaemia National Audit Project (MINAP) [47]. We acknowledge that case ascertainment and changes in coding practices may vary by health outcome and that reliance on ICD coding alone may have led to under reporting of some conditions such as dementia and depression. While we were unable to account for this directly, confounding of temporal changes in ascertainment and coding practices is partially captured by inclusion of calendar year. Furthermore, high sensitivity and specificity of comorbidity recording have been reported for HES, in particular for diabetes (97.7% sensitivity, 96.1% specificity [64]). While dementia diagnoses are delayed by approximately 1.6 years in HES versus primary care, case ascertainment is high (85%) [65]. We did not account for severity of hospitalisation for individuals without MI and were unable to distinguish between subtypes of MI; however, we include the full range of admissions without restriction to less severe disease, and specific MI subtype coding criteria were implemented towards the end of our study period, making future stratification possible [66]. We acknowledge the likely under reporting of lifestyle-related risk factors within hospitalisation records, and baseline cardiovascular risk data, which were restricted to within the study period only. These data were therefore provided in summary format only and represent only the extremes of the population distribution. HES does not capture information with regard to secondary preventative medication, and there is currently no national individual-level hospital prescribing database for England for linkage without consent for research [45]. While we could not quantify the impact of medication on post-MI incidence of health outcomes, we do adjust for invasive management of MI. Finally, we acknowledge that the largest proportion of data relate to outcomes within 1 year of MI due to a drop-off in numbers at risk over follow up, but note that there were sufficient data to provide statistically robust estimates of long-term outcomes (>110,000 individuals and 2,400 individuals per outcome at 5 and 9 years, respectively).

The use of nationally representative health record data provided a depth of analyses allowing risk stratification into clinically relevant groups, for many outcomes. While high-quality system wide healthcare databases are increasingly accessible, major barriers remain in (1) timely data access; (2) access to scalable computational facilities to handle size, complexity, and security standards; and (3) stringent data minimisation, which limited the scope to conditions identified a priori, rather than the full breadth of possible outcomes.

Although clearly a public health focus must be the prevention of MI, evidence from our study has implications for clinical care in Cardiology and beyond. We evidence the excess incidence of conditions that are targeted through current secondary preventative guidelines (heart failure), conditions not currently directly included in secondary preventative guidelines (chronic renal failure and cerebrovascular disease), conditions that would benefit from early detection for improved outcomes (severe bleeding and atrial fibrillation), and conditions that have a significant impact on quality of life (depression). While we did not assess impact of secondary preventative medication on outcomes directly, implications of our research in context of previous studies indicate that (1) improved secondary preventative medication for younger individuals in the most socioeconomically deprived group may tackle the high incidence of post-MI heart failure observed among this demographic [53]; and (2) a reduction in the long-term high incidence of major bleeding complications may be achieved through shortened DAPT regimes [54] and longer-term surveillance following MI. High incidences of chronic renal failure, cerebrovascular disease, and peripheral arterial disease following MI suggest opportunity for intensified secondary prevention of shared modifiable risk factors and enhanced post-MI health surveillance to mitigate against increased healthcare usage and premature death. Moreover, screening interventions for the most at risk of post-MI depression

(including individuals who are younger, female, or socioeconomically deprived) should be considered. While the incidence of vascular dementia contributes only a small proportion of the post-MI disease burden, causal links and opportunities for secondary prevention between MI and vascular dementia warrant further study, given the excess incidence observed.

When extrapolated to the 1.4 million survivors of MI in the UK in 2022, our study implies an estimated 414,400 new diagnoses of heart failure, 312,200 atrial fibrillation, 175,000 cerebrovascular disease, peripheral vascular disease, 266,000 severe bleeding, 380,800 renal failure, 238,000 diabetes mellitus, 109,200 dementia, 124,600 depression, and 189,000 cancer in the next decade, in addition to 99,400 individuals with subsequent MI and up to 529,200 dying within 9 years of first MI.

Our sociodemographic stratified risk charts provide a crucial step in translating future health outcomes to support informed and shared healthcare decision-making. Effective communication of the likely course of disease and risk of adverse long-term outcomes between individuals and healthcare professionals promote positive lifestyle changes, facilitate treatment compliance, and improve patient understanding and quality of life [9,10]. Informed by PPIE, our graphics have been designed in an easy-to-use format via a publicly accessible website, providing healthcare professionals and patients with a tool to discuss relevant demographic-specific risk to direct appropriate care. Moreover, these data have the potential to underpin public health policies aimed at reducing the health inequalities observed and reducing the significant ongoing burden of disease for the increasing number of survivors of MI—some of whom have many potential years of life left. Future work should focus on stratifying risk by specific MI phenotype and identifying modifiable risk factors associated with the increased burden of health outcomes evidenced.

In conclusion, individuals frequently accrue major comorbidities across a range of body systems in the decade following MI—with 3 in 10 developing heart failure or renal failure and 4 in 10 dying. Health inequalities relating to age, sex, and socioeconomic deprivation are clearly evidenced—socioeconomically deprived individuals are more likely to have MI earlier in their life course and experience an increased burden of post-MI health outcomes at an earlier age. Improved post-MI preventative strategies, encompassing enhanced surveillance and detection, are required to tackle the high incidences of heart failure, atrial fibrillation, cerebrovascular disease, and renal failure observed in this population. Finally, sociodemographic stratified risk charts should be used to inform decision-making about health and well-being for specific patient groups in the post-MI period and underpin public health policies aimed at reducing health inequalities.

## Supporting information

**S1 Checklist. REporting of studies Conducted using Observational Routinely collected Data (RECORD) Standard.**
(DOCX)

**S2 Checklist. CODE-EHR framework: Best practice checklist to report on the use of structured electronic healthcare records in clinical research date of completion: 20 January 2023.** Study name: *Health outcomes after myocardial infarction*: *A population study of 56 million people in England.*
(DOCX)

**S1 Text. Review of the prior evidence.**
(DOCX)

**S2 Text. Sensitivity analyses methods.**
(DOCX)

**S3 Text. Hospital Episode Statistics (HES) data cleaning.**
(DOCX)

**S1 Table. Summary of the available evidence of post-MI new onset disease incidence, 1946–October 2023.** ACEi/ARB, angiotensin-converting enzyme inhibitors and angiotensin II receptor blockers; ACS, acute coronary syndromes; AF, atrial fibrillation; CAD, coronary artery disease; CIF, cumulative incidence function; COPD, coronary obstructive pulmonary disease; CPRD, Clinical Practice Research Database; EHR, electronic healthcare record; HES, Hospital Episode Statistics; HF, heart failure; HFrEF, heart failure with reduced ejection fraction; HR, hazard ratio; IRR, incidence rate ratio; IQR, interquartile range; KM, Kaplan–Meier; MI, myocardial infarction; MINAP, Myocardial Ischaemia National Audit Project; NSETMI, non ST-elevation myocardial infarction; PCI, percutaneous coronary intervention; PH, proportional hazards; PPCI, primary percutaneous coronary intervention; PTSD, posttraumatic stress disorder; SCAD, spontaneous coronary artery dissection; SD, standard deviation; STEMI, ST-elevation myocardial infarction; USA, United States of America; vs., versus.
(DOCX)

**S2 Table. Code definitions for health outcomes, vascular risk factors, and invasive coronary strategy for MI according to the International Classification of Diseases (ICD10) and operating Procedure Code Supplement Classification of Interventions and Procedures (OPCS4.5).** ICD10 and OPCS Coding lists adapted from published: https://www.caliberresearch.org/portal. NEC, not elsewhere classifiable; NOC, not otherwise specified.
(DOCX)

**S3 Table. Crude rate of post-MI disease per 1,000 person years (pyrs) for those with MI compared with a matched control group[a] in England, 2008–2017.** [a]Individuals were matched according to single year of age, sex, month and year of hospital admission, and NHS Trust using a 5:1 risk-set matching approach. [b]Cases within the matched control cohort who went on to develop MI were censored at time of first MI; therefore, estimates of subsequent MI for this cohort were not included. CI, confidence interval; MI, myocardial infarction; NA, not applicable; NHS, National Health Service.
(DOCX)

**S4 Table. Crude rate of post-MI disease occurring more than 2 months following study entry (sensitivity analyses) per 1,000 person years (pyrs) for those with MI compared with a matched-control group[a] in England, 2008–2017.** [a]Individuals were matched according to single year of age, sex, month and year of hospital admission, and NHS Trust using a 5:1 risk-set matching approach. [b]Cases within the matched control cohort who went on to develop MI were censored at time of first MI; therefore, estimates of subsequent MI for this cohort were not included. CI, confidence interval; MI, myocardial infarction; NA, not applicable; NHS, National Health Service; SD, standard deviation.
(DOCX)

**S5 Table. Cumulative incidence[a] of all-cause mortality and first hospitalisation for all outcomes following MI treating death without event as a competing risk compared with matched controls[b] at 60 days, 1 year, 5 years, and 9 years of follow-up in England, 2008–2017.** [a]Cumulative incidences are presented as percentage of cases expected to develop each outcome by each respective time point and adjusted for nonlinear age using restricted cubic spline functions, sex, calendar year, and deprivation score—treating death without outcome as

a competing risk. [b]Individuals were matched according to single year of age, sex, month and year of hospital admission, and NHS Trust using a 5:1 risk-set matching approach. [c]Cases within the matched control cohort who went on to develop MI were censored at time of first MI; therefore, estimates of subsequent MI for this cohort were not included. CI, confidence interval; MI, myocardial infarction; NA, not applicable; NHS, National Health Service; SD, standard deviation.

(DOCX)

**S6 Table. Cumulative incidence[a] of all-cause mortality and first hospitalisation for all outcomes at least 2 months following MI (sensitivity analyses) treating death without outcome as a competing risk, compared with matched controls[b] at 1 year, 5 years, and 9 years of follow-up in England, 2008–2017.** [a]Cumulative incidences are presented as percentage of cases expected to develop each outcome by each respective time point and adjusted for nonlinear age using restricted cubic spline functions, sex, calendar year, and deprivation score—treating death without outcome as a competing risk. [b]Individuals were matched according to single year of age, sex, month and year of hospital admission, and NHS Trust using a 5:1 risk-set matching approach. [c]Cases within the matched control cohort who went on to develop MI were censored at time of first MI; therefore, estimates of subsequent MI for this cohort were not included. CI, confidence interval; MI, myocardial infarction; NA, not applicable; NHS, National Health Service; SD, standard deviation.

(DOCX)

**S7 Table. Numbers at risk for the cumulative incidence analysis over time for post-MI outcomes in England, 2008–2017.** [a]Numbers at risk at 1, 5, and 9 years follow-up are equal for those in the main analyses and the sensitivity analyses by design. MI, myocardial infarction.

(DOCX)

**S8 Table. Excess rate[a] of all-cause mortality and first hospitalisation for all outcomes at least 2 months following MI (sensitivity analyses) over and above matched controls[b] in England, 2008–2017.** [a]Excess rate is presented as the aHR for each outcome comparing matched controls with individuals with MI and adjusted for nonlinear age using restricted cubic spline functions, sex, calendar year, and deprivation score—treating death without outcome as a competing risk. [b]Individuals were matched according to single year of age, sex, month and year of hospital admission, and NHS Trust using a 5:1 risk-set matching approach. [c]Cases within the matched control cohort who went on to develop MI were censored at time of first MI; therefore, estimates comparing the HR of subsequent MI between the MI cohort and matched controls are not applicable. aHR, adjusted hazard ratio; CI, confidence interval; MI, myocardial infarction; NA, not applicable.

(DOCX)

**S1 Fig. Adjusted absolute risk[a] over continuous time of subsequent MI, heart failure, atrial fibrillation, cerebrovascular disease, peripheral arterial disease, and severe bleeding occurring at least 2 months following index MI (sensitivity analyses) compared with matched controls[b] in England.** [a]Calculated according to the standardised CIF, treating death without outcome as a competing risk and adjusted for nonlinear age using restricted cubic spline functions, sex, calendar year, and deprivation score. [b]Individuals were matched according to single year of age, sex, month and year of hospital admission, and NHS Trust using a 5:1 risk-set matching approach. CIF, cumulative incidence function; MI, myocardial infarction; NHS, National Health Service.

(TIF)

**S2 Fig. Adjusted absolute risk[a] over continuous time of renal failure, diabetes mellitus, dementia, depression, and cancer occurring at least 2 months following index MI (sensitivity analyses) compared with matched controls[b] in England.** [a]Calculated according to the standardised CIF, treating death without outcome as a competing risk and adjusted for nonlinear age using restricted cubic spline functions, sex, calendar year, and deprivation score. [b]Individuals were matched according to single year of age, sex, month and year of hospital admission, and NHS Trust using a 5:1 risk-set matching approach. [c]Includes all cancer types (ICD10 codes C00–C97), i.e., this category is not restricted to the sum of breast, prostate, lung, and colorectal cancer). CIF, cumulative incidence function; MI, myocardial infarction; NHS, National Health Service.
(TIF)

**S3 Fig. Absolute risk[a] of 11 non-fatal health outcomes occurring at least 2 months following MI at 1 year and 5 years of follow-up (sensitivity analyses) as well as by age group, sex, and deprivation[b] in England ($N$ = 433,361).** [a]Calculated according to the standardised CIF, treating death without outcome as a competing risk and adjusted for nonlinear age using restricted cubic spline functions, sex, calendar year, deprivation score, and receipt of invasive coronary angiography, percutaneous coronary intervention, or coronary artery bypass graft. [b]Deprivation is measured using the IMD where 1 indicates those in the least deprived quintile, and 5 indicates those in the most deprived quintile. Interactive version of these data are provided (https://multimorbidity-research-leeds.github.io/research-resources). CIF, cumulative incidence function; IMD, Index of Multiple Deprivation; MI, myocardial infarction.
(TIF)

**S4 Fig. Absolute risk[a] of 11 non-fatal health outcomes and all-cause mortality at 60 days, 1 year, and 5 years of follow-up by age group, sex, and deprivation[b] among matched individuals without MI in England ($N$ = 2,001,310).** [a]Calculated according to the standardised CIF, treating death without outcome as a competing risk and adjusted for nonlinear age using restricted cubic spline functions, sex, calendar year, and deprivation score. [b]Deprivation is measured using the IMD where 1 indicates those in the least deprived fifth, and 5 indicates those in the most deprived fifth. Interactive version of these data are provided (https://multimorbidity-research-leeds.github.io/research-resources). CIF, cumulative incidence function; IMD, Index of Multiple Deprivation; MI, myocardial infarction.
(TIF)

**S5 Fig. Absolute risk[a] of 11 non-fatal health outcomes occurring at least 2 months following study entry (sensitivity analyses) at 1 year and 5 years of follow-up and by age group, sex, and deprivation[b] among matched individuals without MI in England ($N$ = 2,001,310).** [a]Calculated according to the standardised CIF, treating death without outcome as a competing risk and adjusted for nonlinear age using restricted cubic spline functions, sex, calendar year, and deprivation score. [b]Deprivation is measured using the IMD where 1 indicates those in the least deprived fifth, and 5 indicates those in the most deprived fifth. Interactive version of these data are provided (https://multimorbidity-research-leeds.github.io/research-resources). CIF, cumulative incidence function; IMD, Index of Multiple Deprivation; MI, myocardial infarction.
(TIF)

## Acknowledgments

We acknowledge the work by NHS Digital in providing access to these data for the purposes of our study and the staff in the Leeds Institute for Data Analytics DAT team, University of Leeds

involved in the data management and the secure and safe storage of data for this project. We would like to acknowledge the work by Dr Charlotte Sturley in enhancing our PPIE activities and establishing an ongoing PPIE group for individuals with cardiovascular disease and multiple long-term conditions to support future research, and that of Heart Voices for their help in disseminating our PPIE opportunities. Finally, we would like to acknowledge all the patients and carers who have taken the time to share their experiences of post-heart attack care for their valuable contributions to setting the direction and design of the research, their feedback, and their ongoing involvement in dissemination of the work.

## Author Contributions

**Conceptualization:** Marlous Hall, Paul C. Lambert, Harry Hemingway, Chris P. Gale.

**Data curation:** Marlous Hall.

**Formal analysis:** Marlous Hall, Lesley Smith, Jianhua Wu, Chris Hayward, Jonathan A. Batty.

**Funding acquisition:** Marlous Hall.

**Investigation:** Marlous Hall.

**Methodology:** Marlous Hall, Lesley Smith, Jianhua Wu, Jonathan A. Batty, Paul C. Lambert, Harry Hemingway.

**Project administration:** Marlous Hall.

**Software:** Chris Hayward.

**Supervision:** Jianhua Wu, Paul C. Lambert, Harry Hemingway, Chris P. Gale.

**Validation:** Marlous Hall.

**Visualization:** Marlous Hall, Chris Hayward, Jonathan A. Batty.

**Writing – original draft:** Marlous Hall, Lesley Smith.

**Writing – review & editing:** Marlous Hall, Lesley Smith, Jianhua Wu, Chris Hayward, Jonathan A. Batty, Paul C. Lambert, Harry Hemingway, Chris P. Gale.

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
