## [Editor Report · Decision Letter 0]

29 Jun 2023

Dear Dr Hall, 

Thank you for submitting your manuscript entitled "Health outcomes after myocardial infarction: a population study of 56 million people in England" for consideration by PLOS Medicine.

Your manuscript has now been evaluated by the PLOS Medicine editorial staff and I am writing to let you know that we would like to send your submission out for external peer review.

Please re-submit your manuscript within two working days, i.e. by Jul 03 2023 11:59PM.

Kind regards,

Alexandra Schaefer, PhD

Associate Editor

PLOS Medicine

---

## [Decision Letter · Decision Letter 1]

14 Sep 2023

Dear Dr. Hall,

Thank you very much for submitting your manuscript "Health outcomes after myocardial infarction: a population study of 56 million people in England" (PMEDICINE-D-23-01789R1) for consideration at PLOS Medicine. We would also like to thank you for your prompt response to our enquiry regarding your submission.

Your paper was evaluated by an associate editor and discussed among all the editors here. It was also discussed with an academic editor with relevant expertise, and sent to independent reviewers, including a statistical reviewer. The reviews are appended at the bottom of this email and any accompanying reviewer attachments can be seen via the link below:

[LINK]

In light of these reviews, I am afraid that we will not be able to accept the manuscript for publication in the journal in its current form, but we would like to consider a revised version that addresses the reviewers' and editors' comments. Obviously we cannot make any decision about publication until we have seen the revised manuscript and your response, and we plan to seek re-review by one or more of the reviewers. 

We expect to receive your revised manuscript by Oct 05 2023 11:59PM. Please email me (aschaefer@plos.org) if you have any questions or concerns.

We look forward to receiving your revised manuscript. 

Sincerely,

Alexandra Schaefer, PhD

PLOS Medicine

plosmedicine.org

GENERAL COMMENTS

Please respond to all editor and reviewer comments.

Please cite the reference numbers in square brackets (e.g., “We used the techniques developed by our colleagues [19] to analyze the data”). Citations should be preceding punctuation.

Please cite your Supporting Information as outlined here: https://journals.plos.org/plosmedicine/s/supporting-information

We have noticed that you have used many different footnote markers in your tables and figures. We suggest that you use a consistent format and not mix footnote markers in the same figure/table.

The Data sharing statement (ll.430-454) should be included in the Data Availability section of the manuscript submission form.

The Competing Interests section (ll.401-411) and the Role of the funding source (ll.412-421) should be included in the corresponding sections of the manuscript submission form.

Please remove the ‘Transparency statement’ from the manuscript.

For all observational studies, in the manuscript text, please indicate: (1) the specific hypotheses you intended to test, (2) the analytical methods by which you planned to test them, (3) the analyses you actually performed, and (4) when reported analyses differ from those that were planned, transparent explanations for differences that affect the reliability of the study's results. If a reported analysis was performed based on an interesting but unanticipated pattern in the data, please be clear that the analysis was data-driven.

Did your study have a prospective protocol or analysis plan? Please state this (either way) early in the Methods section.

EDITOR-IN-CHIEF COMMENTS

The manuscript in its current form does not seem to be well placed in the current literature, and it is also crucial to discuss what the study adds to existing research and how the results compare to previous research. In addition, propensity score matching should be considered for better matching of individuals.

ABSTRACT

Abstract Methods and Findings:

* Please provide brief demographic details of the study population (e.g. sex, age).

* Please include the main outcome measures.

Abstract Conclusions:

* Please address the study implications without overreaching what can be concluded from the data; the phrase "In this study, we observed ..." may be useful.

* Please interpret the study based on the results presented in the abstract, emphasizing what is new without overstating your conclusions.

* Please avoid vague statements such as "these results have major implications for policy/clinical care" and please avoid assertions of primacy ("We report for the first time...."). Mention only specific implications substantiated by the results.

Please ensure that all numbers presented in the abstract are present and identical to numbers presented in the main manuscript text.

PLOS Medicine requests that main results are quantified with 95% CIs as well as p values. When reporting p values please report as p<0.001 and where higher as the exact p value p=0.002, for example. For the purposes of transparent data reporting, if not including the aforementioned please clearly state the reasons why not.

Please include any important dependent variables that are adjusted for in the analyses.

Throughout, suggest reporting statistical information as follows to improve clarity for the reader “22% (95% CI [13%,28%]; p</=)”. Please amend throughout the abstract and main manuscript.

Please note the use of commas to separate upper and lower bounds, as opposed to hyphens as these can be confused with reporting of negative values.

When a p value is given, please specify the statistical test used to determine it.

l.48: Please define ‘SD’ at first use and ensure to define all statistical abbreviations at first use in the abstract.

AUTHOR SUMMARY

Thank you for providing a ‘Research in Context’ section. At this stage, we ask you to exchange this section with a short, non-technical Author Summary of your research to make findings accessible to a wide audience that includes both scientists and non-scientists. The Author Summary should immediately follow the Abstract in your revised manuscript. This text is subject to editorial change and should be distinct from the scientific abstract. Please see our author guidelines for more information: https://journals.plos.org/plosmedicine/s/revising-your-manuscript#loc-author-summary.

The summary should include 2-3 single sentence, individual bullet points under each of the questions. The last bullet point should describe the main limitation(s) of the study's methodology.

It may be helpful to review currently published articles for examples which can be found on our website here https://journals.plos.org/plosmedicine/

INTRODUCTION

Please sufficiently address past research and explain the need for and potential importance of your study. Indicate whether your study is novel and how you determined that. If there has been a systematic review of the evidence related to your study (or you have conducted one), please refer to and reference that review and indicate whether it supports the need for your study. 

Please conclude the Introduction with a clear description of the study question or hypothesis.

l.100: Please define ‘US’ at first use. The reference is better placed at the end of the statement.

ll.120-123: Please use the past tense to describe what you did in your study.

METHODS AND RESULTS 

Please ensure that the study is reported according to the RECORD guideline and include the completed RECORD checklist as Supporting Information. You may also include the CODE-EHR checklist. When completing the checklist, please use section and paragraph numbers, rather than page numbers. 

Please add the following statement, or similar (adjust ll.204-205), to the Methods: "This study is reported as per the REporting of studies Conducted using Observational Routinely-collected Data (RECORD) guideline (S1 Checklist)."

PLOS Medicine requests that main results are quantified with 95% CIs as well as p values. When reporting p values please report as p<0.001 and where higher as the exact p value p=0.002, for example. For the purposes of transparent data reporting, if not including the aforementioned please clearly state the reasons why not.

Please include any important dependent variables that are adjusted for in the analyses.

Suggest reporting statistical information as detailed above – see under ABSTRACT

Please present numerators and denominators for percentages, at least in the Tables [not necessarily each time they're mentioned].

l.126: Is it ‘all individuals aged >18 years’ or ‘all individuals aged ≥18 years’?

l.145: Please define ‘HDR’ and ‘UK’.

l.177: Please use a consistent format when writing ‘post MI’/’post-MI’. Please revise throughout the entire manuscript.

ll.181-182, please change to: “Each model was adjusted for age, sex, calendar year, deprivation score and a time-varying covariate for cohort.”

l.191: Please revise the definition of age groups, e.g. 40 years old should not be included in two age groups (i.e. ≤40 and 40 to ≤50). 

l.220: Please define ‘PPIE’ at first use.

l.231: Please define ‘GP’ at first use.

l.246: We suggest adding the numbers for ‘minimal differences in deprivation between cohorts’.

l.253: For easy comparison, we suggest adding the numbers for the matched controls. Please report numerical results together with their statistical information whenever possible – please revise throughout the manuscript.

l.277: Please add a right parenthesis following ‘(9.0% [8.9─9.1%] vs. 8.8 [8.8─8.9%]’.

l.277/278/279: Please add a unit (here, %) when necessary (e.g., for 8.8, 1.79, 8.34) - – please revise throughout the manuscript.

l.284: Given the broad and global audience, we suggest adding more information about the Index of Multiple Deprivation in the Methods section, for example, explaining the meaning of the quintiles.

ll.289-291: Please revise the statement. Your study is observational and therefore causality cannot be inferred. Please remove language that implies causality, such as ‘effect’. Refer to associations instead.

l.302: Please also include the link to the interactive version to the Data availability statement in the manuscript submission form.

DISCUSSION

The discussion in its present form does not seem to be of sufficient depth. Please present and organize the Discussion as follows: a short, clear summary of the article's findings; what the study adds to existing research and where and why the results may differ from previous research; strengths and limitations of the study; implications and next steps for research, clinical practice, and/or public policy; one-paragraph conclusion. Please remove any subheadings.

TABLES

Please note the use of commas to separate upper and lower bounds, as opposed to hyphens as these can be confused with reporting of negative values.

Please define abbreviations used in the tables (including those in Supporting Information files).

Table 1: Please add the unit ‘years’ for ‘Age at study entry (mean, SD)’.

Table 1: We suggest changing ‘Sex (N, % Male)’ to ‘Male Sex (N, %)’.

Table 1: For ‘Socioeconomic Status’, we suggest adding a footnote explaining that this is according to the Index of Multiple Deprivation.

Table 1: Please define ‘KM’, ‘ICD’.

Table 1: For consistency, please use the term ‘3-digit’ throughout the table and the table description. 

FIGURES

For all Figures, please ensure that you have complied with our figures requirements http://journals.plos.org/plosmedicine/s/figures.

Please define abbreviations used in the figure legend of each figure (including those in Supporting Information files).

Please consider avoiding the use of red and green in order to make your figure more accessible to those with colour blindness (including those in Supporting Information files).

Figure 1: Please define ‘MI’, ‘HES’, ‘NHS’.

Figure 1: The footnotes for ‘$$$’ and ‘**’ are missing.

Figure 1: Due to the size of the fonts, the information provided in the boxes is difficult to read and the footnotes are not visible. 

Figure 1: We suggest adding details about the matched cohort in the figure description (i.e., What are the matching factors?).

Figure 2: Please add a unit for age. Please change ‘sd’ to ‘SD’. Please define ‘MI’, ‘NA’, ‘SD’, ‘vs’. 

Figure 2: Could the crude incidence be presented as a Supporting Information file? The figure contains a lot of information and could be confusing for the reader. If you consider the information on the crude incidence to be central to the understanding of the paper, we suggest splitting this figure into a table showing the crude incidence and a figure showing the excess rate.

Figure 3: Please be consistent describing numbers in words or numerals.

Figure 3: The graphs are much too small. Please increase their sizes. 

Figure 3: Please change ‘*includes all cancer […]’ to ‘*includes all cancer types […]’.

Figure 3: We suggest changing the graphs so that the y-axis is identical for all figures to facilitate comparison.

Figure 3: In the description it is suggested that the 95% CIs are presented as shaded areas, but none are visible. Is this correct? 

Figure 3: We suggest adding a definition of the 11 non-fatal health outcomes and 9 key subgroups to the figure description.

Figure 3: Please define ‘MI’, ‘ICD’.

Figure 4: Please define ‘MI’.

Figure 4: Please add a header or similar for the description of time points (60 days Male etc.).

Figure 4: Where does the asterisk belong to?

SUPPLEMENTARY MATERIAL

For references provided in the Supplementary Material – see under REFERENCES.

Methods S1/S2/S3: Please define all abbreviations at first use including the titles (e.g., ‘MI’ or ‘US’).

Table S1: Please thoroughly check for any abbreviations used in the table and define these in or underneath the table. Please write ‘SD’ instead of ‘sd’.

Table S1: For Jernberg 2015, are these 95% CI values presented in ‘Post-MI disease incidence results’? Please ensure to define the statistical details in parenthesis. Please revise throughout the entire table.

Table S1: For Roe 2013, please add a unit for age (column ‘Cohort details’). Please revise throughout the entire table.

Table S1: For Grodzinsky 2015, please add a square bracket open in ‘CAD IRR = 0.89, 95% CI 0.77–1.02]’.

Table S2: Please define ‘ICD’, ‘OPCS’, ‘NEC’, ‘NOC’.

Table S3: Please add a unit for age. Please change ‘sd’ to ‘SD’. Please define ‘CI’, ‘MI’, ‘SD’. 

Table S3/S4/S5/S7: We suggest adding details about the matched cohort in the table description (i.e., What are the matching factors?).

Table S4: Please define ‘CI’, ‘MI’, ‘NA’. 

Table S5: Please define ‘CI’, ‘MI’, ‘NA’. 

Table S5: Please define ‘MI’. Please use a consistent number format (2994 versus 3,032).

Table S7: Please define ‘NA’.

You have used different descriptions for ‘death without disease’ including ‘death without outcome’ or ‘death without event’. For clarity, please use a consistent description throughout the Supplementary Tables.

Figure S1: Where is footnote ‘a’? There is no figure description. The lines in the graphs are dashed but the legend depicts solid lines – please revise. Please see comments for Figure 3 and revise accordingly.

Figure S2/S4: Please define ‘MI’. Please add a header or similar for the description of time points as it is not clear what they refer to (60 days Male, 1 year Male etc.). We suggest adding details about the matched cohort in the figure description (i.e., What are the matching factors?).

Figure S3: Please define ‘MI’. Please add a header or similar for the description of time points as it is not clear what they refer to (60 days Male etc.)

REFERENCES

PLOS uses the numbered citation (citation-sequence) method and first six authors, et al.

Please ensure that journal name abbreviations match those found in the National Center for Biotechnology Information (NCBI) databases (http://www.ncbi.nlm.nih.gov/nlmcatalog/journals), and are appropriately formatted and capitalised.

Please also see https://journals.plos.org/plosmedicine/s/submission-guidelines#loc-references for further details on reference formatting. 

Where website addresses are cited, please specify the date of access.

Comments from the reviewers:

Reviewer #1: Health outcomes after myocardial infarction: a population study of 56 million people in England

In this manuscript A/prof Hall and co-workers report on the post- myocardial infarction (MI) outcomes in a vary large nationwide cohort UK. From about 34 millions UK adults contributing to hospital admissions (about 146 millions) from 2008 to 2017 and who were included, about 443K had MI. The incidence of a range of outcomes (fatal and non-fatal) both cardiovascular and non-cardiovascular was high, with many exceeding those observed in a matched non-MI sub-cohort.

The study is novel in the sense that it goes beyond what previous studies have achieved, and is based in a large sample, therefore translating into stable estimates of the outcomes considered; and as such has a potential to add to the existing body of knowledge on the outcomes of people experiencing an MI, a leading cause of deaths worldwide. The authors should also be commended on the application of robust analytic methods. The study also has some limitations or points to be addressed to get the right message across.

1) The authors are suggesting that the only criteria they used to exclude participants with prevalent outcome of interest at baseline was prior hospitalization for the outcome of interest (see line 257-359). Should this be the case, then it is rather a weak criteria and many people diagnosed with the outcome of interest during follow would actually be people who had the outcome already at baseline and that had not yet caused a hospitalization in the past; translating into an inflation of the incidence. Taking the case of diabetes for illustration, at the average age of participants at inclusion (67 years), diabetes prevalence would be in the region of 10-20% in the general population and likely even higher in those with MI. One query to the authors would be to know if the electronic database didn't include other indicators of existing comorbidities, they use to exclude those with known outcomes at baseline to improve the estimates of their incident outcomes during follow-up.

2) With vascular outcomes (expectedly) dominating the outcomes during follow-up, it is a severe limitation that the description of the baseline profile of the participants does not include indicators of their vascular risk profiles: smoking, hypertension, dyslipidemia, diabetes etc. Should this data be available in the database, then it should be presented with related treatments.

3) Linked to #2, the outcomes of patients with MI are assumed to be influenced by the acute treatment as well as implementation of secondary prevention therapies following the events. This paper cannot afford not to report on acute treatment of the MI (revascularization) and implementation/uptake of secondary prevention therapies at baseline; then testing their effect on the incidence of the outcomes (vascular in particular during follow-up). It is only after achieving this that they can speculate as they have done on lines 311-313 on the importance of intensifying secondary prevention.

Reviewer #2: 

This article is on the incidence of a range of health outcome following an MI, as determined from record linkage of hospitalisations in England. The report uses a large data set and the survival analysis generally seems appropriate, but there are nonetheless some areas that are not entirely clear:

- The event rates following on from MI may be informative, but the control group against which outcomes for MI is compared involves all other participants entering hospital for any reason. It's therefore not surprising that there are differences compared to the MI group, but given the wide variety and severity of other hospitalisations it's difficult to interpret what this means in practice. Similar results could presumably be seen with any more severe hospitalisation event like an MI against a collection of other events including those which may be much less serious. Although there is also a matched cohort, using "age, sex, month and year of hospital admission, and NHS Trust", this would also not account for the different severity of admissions and what meaningful interpretation can be made against the MI specific group. 

- The explanation for why there is such a substantial decrease in risk of cancers is odd, as if aspirin played such a significant role then presumably it could/would be widely used to prevent a huge number of events throughout the general population. Potentially it could have something to do with the inclusion/exclusion criteria of events for the study, but given the scale of the finding it would be useful to have this commented on further as will also place into context the strength of association for where MI increases risk.

- The analysis as described in line 182 suggests that there is a "time varying covariate for cohort", but it isn't entirely clear what is meant by this and if it is actually a time-updated survival analysis being used. If the cohorts relate to the MI and matched groups then not sure how or why these would change over time following initial classification. If the cohorts refer to the combinations of age, sex and socioeconomic deprivation then it is possible that age or deprivation can change over time, but with the follow up of at most 9 years then would presumably not be much change in these. 

- The conclusions in both the abstract and main text that "four in ten [patients] die within ten years" following an MI should probably be placed in context of the age of the population (~67) and that is only slightly higher (39.6%) compared to the age matched hospital population (37.3%), as would seem logical that deaths will occur within an older population irrespective of having an MI. Also while technically true, the abstract refers to "within ten years" whereas table 1 suggests the total follow up period is "9 years". 

- Numbers at risk in table S6 drop quickly from the initial totals, with ~25% not included by year 1, 66% by year 5 and ~99% by the full 9 years of potential follow up between 2008-2017. The cumulative incidence within figure 3 also suggests that majority of excess risk is within first year post hospitalisation and then follows reasonably consistent pattern between groups after this. Within the article not much comment seems to be made on this, but this affects the interpretation of long term MI outcomes given most of the event data actually seems to relate to occurrences within a year of primary MI. 

- I'm not sure if the analysis with events excluded if they occur within two months is technically a landmark analysis, as it isn't being used to classify "responders" after a certain time. Instead it is just excluding from consideration/totals all events in the first two months after hospitalisation. There is some justification for doing this as described by the authors, but is then a bit confusing that this is referred to throughout but only appears in supplementary analysis as main results include events within 30 days.

- In the matched group, it is stated that censoring is done at the time of any future MI. Given the large number of events associated with MI as seen with the first two months after primary event, this could potentially exclude some other events from the matched group and inflate the differences compared to the MI group. Even if it is considered necessary for some analysis, the numbers within the matched group with subsequent MI is presumably known and could be reported for comparison. 

- Figure 1 suggests there are ~9.3 million duplicated records that get excluded. Although numbers will be large simply due to nature of study, and are also still left with large numbers after exclusion, the process behind what generated the duplicates may be relevant to note if it is simply an administrative process, or instead if this could represent any concerns/bias with the data. 

Reviewer #3: see attachment

Reviewer #4: Review of the paper entitled "Health outcomes after MI: a population study of 56 million people in England" by Hall et al. 

This paper reports on a nationwide cohort study including all individuals 18 or older admitted to NHS facilities in England between 2008 and 2017. The analysis pertains to 430 3361 MI. Reliance on the electronic medical record system in place through the NHS enable the authors to report on death and on 11 non-fatal outcomes after MI including both cardiovascular and non-cardiovascular events (including heart failure, renal failure, atrial fibrillation, bleeding, diabetes, cancer, cerebro-vascular disease depression, dementia, recurrent MI, peripheral arterial disease). MIs and outcomes were ascertained by ICD 10 codes. The authors compared these outcomes with those observed in a matched population and by doing so are able to document an excess risk of mortality and morbidity except for cancer and dementia.

The paper includes a number of important strengths: 

* size of the data set

* ability, through the comprehensive electronic medical record system, to report on holistic outcomes beyond the cardiovascular system. 

* comparison to matched controls 

* Information on deprivation

Several shortcomings hinder the enthusiasm for the manuscript. These include in particular:

* The need to clarify the objectives of the paper in order to bring a sharper focus to the incremental knowledge that it will bring to existing literature. There is abundant data on cardiovascular events after MI included but not limited to the work published by the groups of Krumholz (cited only twice) Roger and Goldberg (not cited) . The relatively recent state-of-the-art review by Bahit et al (2018 J Am Coll Cardiol HF. 2018 Mar, 6 (3) 179-186 provides the useful summary of the burden of heart failure after MI. Other references that are useful to consider include among many the following:

o Temporal Trends in Post Myocardial Infarction Heart Failure and Outcomes Among Older Adults.Kochar A, Doll JA, Liang L, Curran J, Peterson ED.J Card Fail. 2022 Apr;28(4):531-539. doi: 10.1016/j.cardfail.2021.09.001. Epub 2021 Oct 5.PMID: 34624511

o Long-Term Mortality of Older Patients With Acute Myocardial Infarction Treated in US Clinical Practice. Kochar A, Chen AY, Sharma PP, Pagidipati NJ, Fonarow GC, Cowper PA, Roe MT, Peterson ED, Wang TY.J Am Heart Assoc. 2018 Jun 30;7(13):e007230. doi: 10.1161/JAHA.117.007230.PMID: 2996099

* A more comprehensive review of the existing literature will help the authors identify the unique contributions of their work and will help them leverage the strengths mentioned above. In particular and Contrasting with the statement on Page 3, some of these studies present cumulative incidence data, account for confounding, and for competing risk of death.

* The ascertainment of both MI and the 11 non-fatal outcomes rely on ICD 10 codes. The limitations of reliance on codes particularly over an extended period of time should be acknowledged as coding practices may have evolved over time. 

* Treatment patterns are not discussed. Have they changed during that time.? Were they adjusted for in the analysis?

* While clearly the burden of cardiovascular and non-cardiovascular events after MI is striking particularly when presented on a national scale, the authors state on page 15 that "new strategies encompassing enhanced surveillance and detection, are required to address the latent burden of new cases of diseases associated with MI". Such vague and sweeping statements are not particularly helpful. How do the authors envision to implement and deploy the strategies that they are advocating for? The same consideration applied to the statement on page 16 lines 366 to 368.

* The similar risk of dementia among cases as compared to controls is somewhat surprising given the association between cardiovascular diseases and dementia and the excess risk of dementia among people with cardiovascular disease. This raises a concern about ascertainment issues and deserves a comment. 

* Reliance on hospitalized events for outcome ascertainment introduce a significant degree of bias which may apply differentially depending on the outcomes under consideration. Reliance of hospitalization is particularly problematic for cancers and dementia events that were not significantly increased after MI in the data set.

* The discussion is difficult to follow and would benefit from adding sub-headings and focusing for example on cardiovascular outcomes first followed by non-cardiovascular events etc. Some of the statements about secondary prevention and the risk of bleeding are not supported by the data. Are there any data on the use of dual antiplatelet therapy in that cohort? Did the risk of bleeding decrease overtime as dual antiplatelet regimens were altered?.

[LINK]

---

## [Decision Letter · Decision Letter 2]

18 Dec 2023

Dear Dr. Hall,

Thank you very much for re-submitting your manuscript "Health outcomes after myocardial infarction: a population study of 56 million people in England" (PMEDICINE-D-23-01789R2) for review by PLOS Medicine.

Thank you for your detailed response to the editors' and reviewers' comments. I have discussed the paper with my colleagues and the academic editor, and it has also been seen again by two of the original reviewers. The changes made to the paper were mostly satisfactory to the reviewers. As such, we intend to accept the paper for publication, pending your attention to the editorial comments below in a final revision. When submitting your revised paper, please once again include a detailed point-by-point response to the editorial comments. Please ensure that the response letter indicates the location (lines and pages) of the changes made in the manuscript.

[LINK]

We expect to receive your revised manuscript within 1 week. Please email me (aschaefer@plos.org) if you have any questions or concerns.

We look forward to receiving the revised manuscript by Dec 25 2023 11:59PM.   

Sincerely,

Alexandra Schaefer, PhD

Associate Editor 

PLOS Medicine

plosmedicine.org

Requests from Editors:

ACADEMIC EDITOR COMMENTS

I have gone through the answers of the authors and revised manuscript. They have done what was in their capacity to address my comments (and I believe those of other reviewers) and made efforts to acknowledge the remaining limitations. The length of the manuscript is justified at least in part by the density of the comments raised by the four reviewers. Having to address them has required from the authors to add substantial new text.

EDITORIAL POINTS

1) While we acknowledge that new text has been added to the revised version of the manuscript since the last revision, the editors agree with reviewer #4 that the text, particularly the discussion, needs to emphasize the key messages more clearly and thus be condensed to improve the readability of the manuscript. In addition, we believe that the reliance on ICD codes and the consequent under-reporting of some comorbid conditions, particularly dementia, should be discussed more clearly as a limitation.

2) Please be consistent in your writing of “long-term/short-term” or “long term/short term”.

3) Please remove any units from brackets presenting 95% CI values. As the preceding number is presented with a unit, it sufficient to present the 95% CI values as numbers only.

AUTHOR SUMMARY

Thank you for providing the Author summary. The current summary is rather long. The bullet points should reflect the core messages/findings for each question. You should aim for 2-3 bullet points for each question with a single sentence per bullet point. Please use non-technical language.

INTRODUCTION

1) l.143: Please add “To our knowledge, there were no…”

2) l.147: Please add “To our knowledge, published articles to date have..”

3) ll.166-171: There is no need to list all health outcomes in full more than once in the paragraph. We suggest shortening this to: “Therefore, we determined the excess incidence, estimated the adjusted absolute risk, and presented age, sex, deprivation and time specific risk charts for each of these health outcomes following MI compared with matched controls.” 

METHODS AND RESULTS

1) l.341: Please define ‘SD’ before first use.

2) ll.379-383: Please ensure that CI values are presented in square brackets and that before each set of square brackets “95% CI” is added.

3) ll.382 and ff: “There was an excess rate of heart failure (adjusted HR 4.93; 95% CI 4.89,7.97;p<0.001)…”. In the following brackets, you only refer to “HR”. Are the values that follow for the other outcomes also adjusted hazard ratios? We feel the current presentation might be confusing to readers. We suggest introducing the abbreviation "aHR" on line 267 and using "aHR" from then on. Please revise.

4) Figure 2: Please change the x-axis label and the column header to “adjusted Hazard Ratio”. You have already defined ‘CI’ in the figure description (“…and 95% confidence intervals (CIs)..”); please remove the definition of ‘CI’ below the description.

DISCUSSION

Please remove the ‘Conclusion’ heading. The conclusion paragraph should be part of a continuous discussion.

REFERENCES

When specifying the date of access, please write “accessed” instead of “Date accessed”.

SUPPLEMENTARY MATERIAL

1) S1 and S2 Checklist contain the same checklist. Please check and provide both, the RECORD checklist and the CODE-EHR checklist.

2) S2 Figure: The definition of footnote c is missing in the figure description.

3) S2 Table: You have already defined ‘ICD’ and ‘OPCS’ in the table title, i.e. you can remove these two definitions from the definitions listed underneath.

3) S3/S4 Table: Similarly, to S2 Table, please note that you only need to define abbreviations once, either in the title or in the definitions below the table (also applies for figures). Please revise throughout the entire main manuscript and the supporting files.

SOCIAL MEDIA

To help us extend the reach of your research, please provide any X (formerly known as Twitter) handle(s) that would be appropriate to tag, including your own, your coauthors’, your institution, funder, or lab. Please respond to this email with any handles you wish to be included when we tweet this paper.

Comments from Reviewers:

Reviewer #2: 

Reviewer 2

I appreciate the authors detailed responses to the issues previously raised and believe that the majority of these have been resolved. 

However, for comment 3 although the text has been changed to "time dependent effect for cohort" and this reflects the terminology used in the referenced paper for the standsurv function, I think this is unusual as "time-updated" and "time-dependent" are more often used for the same meaning of coefficients changing over time as opposed to the model estimates . While the aims of approach taken do seem logical given the potential non proportional hazards over time, with using continuous time for the model as opposed to categorical breaks as in the standsurv publication, it isn't obvious at what (if any) time-points these hazard ratios would apply to if the estimates could end up higher/lower at different times during follow up. If the interpretation is just intended as an average or some other meaning if they are not interpretable the same way as a more conventional survival analysis, could be briefly expanded on within methods. 

Reviewer #4: The authors have made a robust effort to address the comments from the reviewers. My main residual comment is the presentation of the data and the writing of the manuscript. In my opinion, the text is too long and the message therefore gets diluted. A deliberate commitment to being more concise would strengthen the message of the manuscript and make it easier for the readership.

[LINK]

---

## [Editor Report · Decision Letter 3]

5 Jan 2024

Dear Dr Hall, 

On behalf of my colleagues and the Academic Editor, Andre-Pascal Kengne, I am pleased to inform you that we have agreed to publish your manuscript "Health outcomes after myocardial infarction: a population study of 56 million people in England" (PMEDICINE-D-23-01789R3) in PLOS Medicine.

I appreciate your thorough responses to the reviewers' and editors' comments throughout the editorial process. We look forward to publishing your manuscript, and editorially there are only a few remaining minor stylistic/presentation points that should be addressed prior to publication. We will carefully check whether the changes have been made. If you have any questions or concerns regarding these final requests, please feel free to contact me at aschaefer@plos.org.

Please see below the minor points that we request you respond to:

1) The S1 and S2 checklists still contain the same checklist (both contain the CODE-EHR checklist). Please review and submit both the RECORD checklist and the CODE-EHR checklist.

2) Figure 2: In the figure description, you refer to Table S8 to provide a supplementary sensitivity analysis in which follow-up was restricted to start at least 2 months after study entry. However, Table S8 shows the excess rate of all-cause mortality and post-MI hospitalization over matched controls in England. Please check carefully throughout the entire manuscript that you have referenced the appropriate tables/figures in your main text and figure/table descriptions.

3) Please reference S7 table in the main text as it is currently unreferenced in the main text.

4) S8 Figure: Please check the placement of the footnotes. Footnote b is added to 'NA' in the table and 'matched controls' in the title, but footnote c is not added anywhere. It appears that footnote c should be added to 'NA'.

PRESS

Sincerely, 

Alexandra Schaefer, PhD 

Associate Editor 

PLOS Medicine